# Variance-Enlarged Poisson Learning for Graph-based Semi-Supervised Learning with Extremely Sparse Labeled Data

**Xiong Zhou**[1], **Xianming Liu**[1]*, **Hao Yu**[2], **Jialiang Wang**[1], **Zeke Xie**[3], **Junjun Jiang**[1], **Xiangyang Ji**[2]

[1]Harbin Institute of Technology   [2]Tsinghua University   [3]Baidu Research

## Abstract

Graph-based semi-supervised learning, particularly in the context of extremely sparse labeled data, often suffers from trivial solutions where label functions tend to be nearly constant across unlabeled data. In this paper, we introduce Variance-enlarged Poisson Learning (VPL), a simple yet powerful framework tailored to alleviate the issues arising from the presence of trivial solutions. VPL incorporates a variance-enlarged regularization term, which induces a Poisson equation specifically for unlabeled data. This intuitive approach increases the dispersion of labels from their average mean, effectively reducing the likelihood of trivial solutions characterized by nearly constant label functions. We subsequently introduce two streamlined algorithms, V-Laplace and V-Poisson, each intricately designed to enhance Laplace and Poisson learning, respectively. Furthermore, we broaden the scope of VPL to encompass graph neural networks, introducing Variance-enlarged Graph Poisson Networks (V-GPN) to facilitate improved label propagation. To achieve a deeper understanding of VPL's behavior, we conduct a comprehensive theoretical exploration in both discrete and variational cases. Our findings elucidate that VPL inherently amplifies the importance of connections within the same class while concurrently tempering those between different classes. We support our claims with extensive experiments, demonstrating the effectiveness of VPL and showcasing its superiority over existing methods. The code is available at https://github.com/hitcszx/VPL.

## 1 Introduction

Semi-supervised learning (SSL) is a classical machine learning paradigm, which is driven by the desire to harness the vast amount of unlabeled data that exists in the real world, alongside the limited availability of labeled data (Fralick, 1967; Vapnik & Chervonenkis, 1974; Blum & Mitchell, 1998; Berthelot et al., 2019). Traditional supervised learning methods heavily rely on labeled examples for training, which can be costly and time-consuming to obtain, especially for complex tasks in domains like computer vision, natural language processing, and healthcare. Semi-supervised learning seeks to bridge this gap by leveraging both labeled and unlabeled data to improve model performance and generalization. By incorporating the vast pool of unlabeled data, semi-supervised learning algorithms aim to extract more meaningful and robust representations, thereby enhancing the scalability, efficiency, and accuracy of machine learning models (Sohn et al., 2020). The emphasis of this paper is on scenarios where the labeled data is exceptionally scarce. This situation is common in many real-world applications, such as medical diagnosis, where obtaining labeled data can be expensive or time-consuming. The challenge lies in effectively utilizing this limited labeled information to improve model performance.

Graph-based methods provide a powerful framework for leveraging the underlying structure or relationships within the data (Zhu, 2006; Anis et al., 2018; Song et al., 2022). By constructing a graph where nodes represent data points and edges denote relationships or similarities, information can be propagated across the graph to make predictions on unlabeled data. This approach is particularly advantageous in scenarios where data points are not independent, but rather have dependencies or

---

*Correspondence to: Xianming Liu <csxm@hit.edu.cn>

connections that can be exploited for learning. In the realm of graph-based SSL, the following optimization problem is considered:

$$\min_u \mathcal{J}(u) \triangleq \sum_{ij} w_{ij} L\left(u(x_i), u(x_j)\right), \tag{1.1}$$

with some specific constraints applied to labeled points. Herein, $w_{ij}$ denotes the edge weight between vertices $x_i$ and $x_j$, and $L$ is the measure of distance between two labels. When $L$ takes on the form of $\ell_p$ distance and the labeled points adhere the boundary conditions $u(x_i) = y_i$ for $i \in [l]$, the problem above is referred to as *p-Laplace learning* (Zhou & Schölkopf, 2005; Bühler & Hein, 2009). This appellation originates from that at $p = 2$ it is initially known as *Laplace learning* or *label propagation* (Zhu et al., 2003). When $L(u(x_i), u(x_i)) = w_{ij}^{p-1} \|u(x_i) - u(x_j)\|_p^p$, the problem in Eq. 1.1 under certain regularity assumptions follows that the limit of the (unique) minimizers of $\mathcal{J}$ (as $p$ grows to infinity) is the lex-minimizer for *Lipschitz learning* (Egger & Huotari, 1990; Aronsson et al., 2004; Kyng et al., 2015). Moreover, many variants of Laplace learning have been proposed with various types of soft label constraints in place of hard constraints (Zhou et al., 2003; Belkin et al., 2004b;a; Kang et al., 2006; Wang et al., 2013; Gong et al., 2016).

However, it is found that, in the context of large graphs or scenarios with extremely sparse labeled data, distances based on graph Laplacians do not conform to the intuition that nodes within the same clusters can be interconnected via many paths of high probability (Nadler et al., 2005). Instead, they tend to converge to non-informative functions, which results in degenerate solutions that take the form of an almost everywhere constant label function, thereby significantly deteriorating the accuracy performance (Nadler et al., 2009; El Alaoui et al., 2016; Calder et al., 2020). This undesirable property is usually attributed to the fact that long random walks tend to forget their starting position, leading to the mixing of random walks (Lovász & Simonovits, 1990). Several alternative methods have been proposed to address this issue, including higher order graph Laplacian distances (Alamgir & Luxburg, 2011; Zhou & Belkin, 2011; Bridle & Zhu, 2013; Kyng et al., 2015; Rios et al., 2019). More recently, Calder et al. (2020) proposed Poisson learning as a different approach to handling labeled data. This method replaces the original boundary conditions in the Laplace equation with the source term in a graph Poisson equation. However, Poisson learning still treats unlabeled data in the form of Laplace equations, which retains the characteristic of a smooth combination primarily influenced by the unlabeled portion, potentially risking convergence towards non-informative predictions.

In this work, we propose a simple yet effective framework, variance-enlarged Poisson learning (VPL), to address the presence of degenerate solutions that tend to be almost constant for unlabeled points. Specifically, we introduce a variance-enlarged regularization term that increases the dispersion of labels away from the mean or average label by enlarging the variance of labels. This leads to the Poisson equation for unlabeled data. Based on the variance-enlarged framework, we propose two specific algorithms, V-Laplace and V-Poisson, to enhance Laplace learning and Poisson learning, respectively. These algorithms are efficient and simple to implement (see Sec. 3.2). We also incorporate VPL into the propagation in graph neural networks and propose variance-enlarged graph Poisson networks (V-GPN), which can improve label propagation at very low label rates. Furthermore, we provide theoretical analysis in both discrete (finite data) and variational (infinite data) cases. In the discrete case, we demonstrate that VPL is equivalent to reducing edge weights and prove that under mild conditions, it amplifies the importance of edge weights connecting vertices within the same class while diminishing the importance of those connecting vertices from different classes. In the variational case, we provide a variational characterization of the performance of VPL with a more general objective incorporating the $\ell_p$ distance ($p \geq 2$). The theoretical analysis is formulated in the asymptotic limit as the number of unlabeled data grows to infinity while the number of labeled data stays constant. We derive the condition under which the inclusion of the variance-enlarged regularization in VPL guarantees the convexity of the variational objective. We prove that the associated optimality conditions result in a partial differential equation for the estimate of labeling functions, referred to as the weighted $p$-Poisson equation. We conduct extensive experiments to demonstrate the effectiveness of VPL, showcasing its superior performance over the state-of-the-arts.

The main contributions of our work are summarized as follows:

- We present variance-enlarged Poisson learning, a simple yet effective framework for tackling the challenges associated with extremely sparse labeled data in graph-based semi-supervised learning. Our extensive experimental results demonstrate the effectiveness and superior performance of VPL over existing methods.

- We propose two specific algorithms, V-Laplace and V-Poisson, which are efficient and simple to implement and achieve better performance than the original Laplace learning and Poisson learning with extremely sparse labeled data. We also consider the problem of classifying nodes in graph-structured data and propose variance-enlarged graph Poisson networks to enlarge the variance of labels in the propagation.

- We provide theoretical analysis in both discrete (finite data) and variational (infinite data) cases. In the discrete case, we show that VPL is equivalent to reducing edge weights, which implicitly amplifies the importance of edge weights within the same class. In the variational case, we characterize the performance of VPL in the asymptotic limit as the number of data grows to infinity. The associated optimality conditions lead to the weighted $p$-Poisson equation.

## 2 PRELIMINARY

Let $\mu(x)$ be an underlying distribution on a compact space $\Omega \subseteq \mathbb{R}^d$ with a smooth boundary. Let $u : \Omega \to \mathcal{Y}$ be the unknown function we wish to estimate. For classification problems, we let $\mathcal{Y} = \{e_1, ..., e_k\}$ $(k \geq 2)$[1], where $e_i$ denotes the one-hot vector whose $i$-th element is 1. For the standard (transductive) semi-supervised learning problem, we have $l$ labeled examples $\{(x_1, y_1), ..., (x_l, y_l)\}$ and $n - l$ unlabeled examples $\{x_{l+1}, ..., x_n\}$ with all examples $x_i$ sampled i.i.d from $\mu(x)$. We let $X = \{x_1, ..., x_n\}$ denote the vertices of a symmetric graph with edge weights $w_{ij} \geq 0$ that indicate the similarity between $x_i$ and $x_j$. The degree of vertex $x_i$ is denoted by $d_i = \sum_{j=1}^{n} w_{ij}$. The goal of graph-based semi-supervised learning is to construct an estimate of labels for all unlabeled examples by leveraging pairwise similarities in the graph.

**Laplace Learning and Poisson Learning.** The main difference between Laplace learning and Poisson learning emerges from their distinct approaches to handling labeled data. In Laplace learning, labeled data is incorporated as boundary conditions, taking the form $u(x_i) = y_i$ for $1 \leq i \leq l$, while Poisson learning replaces the given labels with the assignment of sources and sinks like flow in the graph, thus resulting in the Poisson equation $\mathcal{L}u(x_i) = y_i - \overline{y}$ (with $\overline{y} = \frac{1}{l} \sum_{j=1}^{l} y_j$) for $1 \leq i \leq l$. On the other hand, both methods treat unlabeled data in the same way as follows:

$$\mathcal{L}u(x_i) = 0, \text{ for } l + 1 \leq i \leq n, \tag{2.1}$$

where $\mathcal{L}$ is the unnormalized graph Laplacian given by $\mathcal{L}u(x_i) = \sum_{j=1}^{n} w_{ij}(u(x_i) - u(x_j))$.

## 3 METHODOLOGY

In this section, we present in detail the variance-enlarged framework and two specific algorithms from this framework, namely V-Laplace and V-Poisson. We also propose the variance-enlarged graph Poisson network (V-GPN) for semi-supervised node classification with graph-structured data.

### 3.1 VARIANCE-ENLARGED POISSON LEARNING

We explore the concept of enlarging the variance on labels, aiming to increase the dispersion of labels from their mean or average labels. This approach intuitively alleviates the presence of degenerate solutions that tend to be almost constant for unlabeled points. More precisely, we focus on the optimization objective as follows:

$$\sum_{i=1}^{n} \sum_{j=1}^{n} w_{ij} L(u(x_i), u(x_j)) - \lambda \cdot \text{Var}[u], \tag{3.1}$$

where $\text{Var}[u]$ denotes the variance of these labels. Here, for brevity, we choose not to introduce constraints on the labeled data. In the subsequent sections, we will specifically incorporate the constraints associated with Laplace learning and Poisson learning with respect to labeled data into our framework, while the inclusion of other constraints is also a viable alternative.

---

[1]For binary classification, we usually set $\mathcal{Y} = \{+1, -1\}$.

For the specific form of variance, there are two approaches available: a direct calculation of variance across all data or a weighted formulation based on certain criteria. In this paper, we adopt the latter one that takes into account the degrees of each point, i.e., $\text{Var}[u] = \sum_{i=1}^{n} q_i \|u(x_i) - \overline{u}\|^2$, where we define $\overline{u} = \sum_{i=1}^{n} q_i u(x_i)$ and $q_i = \frac{d_i}{\sum_{j=1}^{n} d_j}$.

When $L(u(x_i), u(x_j)) = \frac{1}{2}\|u(x_i) - u(x_j)\|^2$, compared with Laplace learning and Poisson learning, the variance-enlarged objective induces the Poisson equation for unlabeled data as follows:

$$\mathcal{L}u(x_i) = \lambda q_i \left(u(x_i) - \overline{u}\right), \text{ for } l+1 \leq i \leq n, \tag{3.2}$$

Formally, our proposed approach remains aligned with the principles of Poisson learning, leading us to designate it as *Variance-Enlarged Poisson Learning*.

**Remark.** Variance-enlarged regularization is somewhat similar to entropy minimization (Grandvalet & Bengio, 2004), as entropy minimization assumes a prior that prefers minimal class overlap and helps enlarge the variance of labels. However, variance-enlarged Poisson learning can be better suited for conducting an optimization algorithm than entropy minimization due to its quadratic nature, making it a practical choice for certain scenarios with more concise iterations (cf. Sec. 3.2).

## 3.2 V-LAPLACE AND V-POISSON

**V-Laplace Learning.** We consider the boundary conditions of labeled points in Laplace Learning, which is equivalent to solving the problem:

$$\begin{cases} \mathcal{L}u(x_i) = \lambda q_i(u(x_i) - \overline{u}) & \text{if } l+1 \leq i \leq n \\ u(x_i) = y_i, & \text{if } 1 \leq i \leq l. \end{cases} \tag{3.3}$$

In Algorithm 1, we provide an algorithm to solve the V-Laplace problem in Eq. 3.3, where we clamp the labels for labeled data and only update predictions for unlabeled data.

---

**Algorithm 1** Variance-enlarged Laplace Learning (V-Laplace)

---

1: **Input:** The label matrix $\mathbf{Y}_l \in \mathbb{R}^{l \times k}$, the weight matrix $\mathbf{W}$, and timesteps $T$, and the trade-off parameter $\lambda$.
2: **Output:** $\mathbf{U} \in \mathbb{R}^{u \times k}$ (where $\mathbf{U}_l$ denotes the predicted matrix restricted to labeled rows).
3: $\mathbf{D} \leftarrow \text{diag}(\mathbf{W1})$
4: $\mathbf{L} \leftarrow \mathbf{D} - \mathbf{W}$
5: $\mathbf{U} \leftarrow \mathbf{zeros}(n, k)$
6: **for** $i = 1$ to $T$ **do**
7: $\quad \overline{\mathbf{U}} \leftarrow \mathbf{D}\left(\mathbf{U} - \frac{\mathbf{11}^\top \mathbf{DU}}{\mathbf{1}^\top \mathbf{D1}}\right)$
8: $\quad \mathbf{U} \leftarrow \mathbf{U} + \mathbf{D}^{-1}(\lambda\overline{\mathbf{U}} - \mathbf{LU})$
9: $\quad \mathbf{U}_l = \mathbf{Y}_l$.
10: **end for**

---

**V-Poisson Learning.** We consider the Poisson equation for the labeled points in Poisson Learning, which is equivalent to solving the problem:

$$\begin{cases} \mathcal{L}u(x_i) = \lambda q_i(u(x_i) - \overline{u}) & \text{if } l+1 \leq i \leq n \\ \mathcal{L}u(x_i) = y_i - \overline{y} + \lambda q_i(u(x_i) - \overline{u}), & \text{if } 1 \leq i \leq l, \end{cases} \tag{3.4}$$

satisfying $\sum_{i=1}^{n} d_i u(x_i) = 0$, where $\overline{y} = \frac{1}{l}\sum_{i=1}^{l} y_i$ is the average label vector.

Unlike V-Laplace, which directly adopts constraints on labeled data in Laplace learning, V-Poisson in Algorithm 2 takes a slightly different way to handle labeled data from standard Poisson learning. Here, we also introduce modifications to the labeled data, since it is a crucial step to ensure that the identity $\sum_{i=1}^{n} \mathcal{L}u(x_i) = 0$ always holds. In the following, we gives the foundation for convergence of variance-enlarged Poisson learning through the random walk perspective.

**Theorem 3.1.** *If $\lambda \leq \min_{ij} \frac{w_{ij}}{q_i q_j}$, the graph $G$ with the weight matrix $W' = W - \lambda qq^\top$ (where $q = (q_1, ..., q_n)^\top$) is connected and the Markov chain induced by the random walk is aperiodic, we*

*have: (1) For the iterations $u_{T+1}(x_i) = u_T(x_i) + d_i^{-1}(\lambda q_i(u_T(x_i) - \overline{u}_T) - \mathcal{L}u_T(x_i))$ of V-Laplace learning in Algorithm 1, $u_T$ converges to the solution of the problem in Eq. 3.3; (2) For the iterations $u_{T+1}(x_i) = u_T(x_i) + d_i^{-1}(\sum_{j=1}^{l}(y_j - \overline{y})\mathbb{I}(i = j) + \lambda q_i(u_T(x_i) - \overline{u}_T) - \mathcal{L}u_T(x_i))$ of V-Poisson learning in Algorithm 2, then $u_T \to u$ as $T \to \infty$, where $u$ is the solution of Poisson equation 3.4 satisfying $\sum_{i=1}^{n} d_i u(x_i) = 0$.*

---

**Algorithm 2** Variance-enlarged Poisson Learning (V-Poisson)

---

1: **Input:** The label matrix $\mathbf{Y}_l \in \mathbb{R}^{l \times k}$, the weight matrix $\mathbf{W}$, and timesteps $T$, and the trade-off parameter $\lambda$.
2: **Output:** The predicted matrix $\mathbf{U} \in \mathbb{R}^{n \times k}$
3: $\mathbf{D} \leftarrow \text{diag}(\mathbf{W1})$
4: $\mathbf{L} \leftarrow \mathbf{D} - \mathbf{W}$
5: $\overline{\mathbf{y}} \leftarrow \frac{1}{m}\mathbf{Y}_l\mathbf{1}$
6: $\mathbf{B} \leftarrow [\mathbf{Y}_l - \overline{\mathbf{y}}, \mathbf{zeros}(u, k)]$
7: $\mathbf{U} \leftarrow \mathbf{zeros}(n, k)$
8: **for** $i = 1$ to $T$ **do**
9: $\quad \overline{\mathbf{U}} \leftarrow \mathbf{D}\left(\mathbf{U} - \frac{\mathbf{11}^\top \mathbf{DU}}{\mathbf{1}^\top \mathbf{D1}}\right)$
10: $\quad \mathbf{U} \leftarrow \mathbf{U} + \mathbf{D}^{-1}\left(\mathbf{B}^\top - \mathbf{LU} + \lambda\overline{\mathbf{U}}\right)$
11: **end for**

---

### 3.3 Variance-Enlarged Graph Poisson Networks

In this subsection, we consider the problem of classifying nodes in graph-structured data (Kipf & Welling, 2017; Veličković et al., 2018), where the graph structure is encoded with a neural network model and trained on a supervised loss for an extremely small set of labels.

Inspired by graph attention networks (GAT) (Veličković et al., 2018), we propose variance-enlarged graph Poisson networks (V-GPN) to adaptively capture the importance of the neighbors exerting to the target node via attention mechanism. In this way, the graph information can be gradually refined via network training. To be specific, we calculate the attention coefficients $a_{ij}$ between nodes $x_i$ and $x_j$ as $a_{ij} = \boldsymbol{h}[\mathbf{H}x_i; \mathbf{H}x_j]$, where $\boldsymbol{h}$ constitutes a trainable weight vector and $\mathbf{H}$ is a trainable weight matrix. Subsequently, these attention coefficient $a_{ij}$ are normalized using a softmax function to compute the edge weight matrix $\mathbf{W}$ that is further optimized via network training, that is, $w_{ij} = \frac{\exp(a_{ij})}{\sum_{k \in \mathcal{N}_i} \exp(a_{ik})}$, where $\mathcal{N}_i$ is some neighborhood of node $i$. To enlarge the variance of labels in the propagation, we obtain the output of our proposed V-GPN by the following iterations:

$$\mathbf{U}^{(t)} \leftarrow \mathbf{U}^{(t-1)} + \mathbf{D}^{-1}\left(\lambda\overline{\mathbf{U}}^{(t-1)} - \mathbf{LU}^{(t-1)}\right), \tag{3.5}$$

where $\overline{\mathbf{U}}^{(t-1)}$ is defined as in Algorithms 1 and 2, $\mathbf{D}$ and $\mathbf{L}$ represent the diagonal matrix of degrees and Laplacian matrix based on the attention-based weight matrix $\mathbf{W}$, respectively. Owing to the specifics of the attention calculation, $\mathbf{D}$ simplifies to an identity matrix. Consequently, we can streamline the iteration process more succinctly as $\mathbf{U}^{(t)} \leftarrow \mathbf{WU}^{(t-1)} + \lambda\overline{\mathbf{U}}^{(t-1)}$.

To incorporate the structural information arising from node features, we follow the approach in (Wan et al., 2021), employing a feature transformation module $f_{FT}$ composed of a single perception layer to predict labels based on node features. This module modifies the output at $(T - 3)$-th iteration, giving rise to $\mathbf{U}^{T-3} \leftarrow \mathbf{U}^{T-3} + f_{FT}(\mathbf{X})$, where $T$ denotes the number of iterations.

## 4 Theoretical Understanding And Insights

In this section, we theoretically investigate variance-enlarged Poisson learning, covering both the discrete (finite data) and variational (infinite data) cases. **All proofs can be found in Appendix A.**

## 4.1 THE DISCRETE CASE

For the discrete case with finite data, let us recall the Laplace equation: $\mathcal{L}u(x_i) = 0$, which yields the expression $u(x_i) = \frac{1}{d_i} \sum_{j=1}^{m} w_{ij} u(x_j) = \frac{1}{d_i} \sum_{j=1}^{l} w_{ij} u(x_j) + \frac{1}{d_i} \sum_{j=l+1}^{n} w_{ij} u(x_j)$. This reveals that $u(x_i)$ results from a smooth combination involving all labels and predictions. Consequently, when labeled data is extremely sparse, the result of $u(x_i)$ is primarily shaped by the unlabeled portion $\frac{1}{d_i} \sum_{j=l+1}^{n} w_{ij} u(x_j)$, potentially leading to a convergence toward non-informative predictions. To mitigate the mixing of label propagation, several studies have focused on re-weighting the graph more heavily near labels (Shi et al., 2017; Calder & Slepčev, 2020), thus increasing the influence of labeled data and implicitly down-weighting the importance of unlabeled data.

We establish that variance-enlarged Poisson learning, under mild conditions, implicitly amplifies the importance of edge weights connecting vertices within the same class, while simultaneously diminishing the importance of those connecting vertices from different classes. We demonstrate that the inclusion of $-\mathrm{Var}[u]$ in Eq. 1.1 is equivalent to explicitly reducing all edge weights:

**Lemma 4.1.** *The solution of the Poisson equation in Eq. 3.2 is equivalent to the minimizer of the objective* $\frac{1}{2} \sum_{i=1}^{n} \sum_{j=1}^{n} (w_{ij} - \lambda q_i q_j) \|u(x_i) - u(x_j)\|_2^2$, *which indicates the optimal condition:*

$$u(x_i) = \frac{1}{d_i - \lambda q_i} \left[ \sum_{j=1}^{m} w_{ij} u(x_j) - \lambda q_i \overline{u} \right] = \frac{1}{d_i - \lambda q_i} \sum_{j=1}^{m} (w_{ij} - \lambda q_i q_j) u(x_j). \quad (4.1)$$

This theorem offers a clear and straightforward understanding of the significance of the variance-enlarged regularization term. Moreover, it implies that defining the parameter range $\lambda \leq \min_{ij} \frac{w_{ij}}{q_i q_j}$ is essential for preserving the convexity of the optimization objective.

To better illustrate the implication of reducing edge weights as indicated in Eq. 4.1, we introduce a reasonable and intuitively meaningful assumption as follows:

**Assumption 4.2.** *For each vertex $x_i$ with its label $y_i$, let $s_i^m(k) = \frac{\sum_{j=1}^{m} \mathbb{I}(y_j=k) w_{ij}}{\sum_{j=1}^{m} \mathbb{I}(y_j=k)}$ (where $\mathbb{I}$ is the identity function), we assume that $s_i^n(y_i) > \max_{k \neq y_i} s_i^n(k)$ and $s_i^l(y_i) > \max_{k \neq y_i} s_i^l(k)$.*

Assumption 4.2 consists of two statements for each vertex $x$: (1) The average of edge weights between $x$ and vertices within the same class should exceed the average of edge weights between $x$ and vertices of any other individual class; (2) The average of edge weights between $x$ and labeled vertices within the same class should exceed the average of edge weights between $x$ and labeled vertices of any other individual class. This assumption resembles the commonly-used manifold assumption in which samples located near each other on a low-dimensional manifold are expected to share similar labels, but it differs in its emphasis on a description with a more global notion of class.

Together with Theorem 4.1 and Assumption 4.2, we can easily derive the following proposition:

**Proposition 4.3.** *Under Assumption 4.2, let $b_i^m(k) = \frac{\sum_{j=1}^{m} \mathbb{I}(y_j=k) q_i q_j}{\sum_{j=1}^{m} \mathbb{I}(y_j=k)}$, if for $m \in \{l, n\}$ and $k \in \mathcal{Y}$, we have $\lambda \leq \min_{ij} \frac{w_{ij}}{q_i q_j}$, $b_i^m(y_i) \leq b_i^m(k)$, then $\frac{s_i^m(y_i) - \lambda b_i^m(y_i)}{s_i^m(k) - \lambda b_i^m(k)} > \frac{s_i^m(y_i)}{s_i^m(k)}$.*

**Remark.** This proposition provide the insight that a reduction in edge weights can help give wider influence of vertices within the same class, thus forcing the solution of predictions with larger within-class similarities, *i.e.*, $\frac{s_i^n(y_i) - \lambda b_i^n(y_i)}{s_i^n(k) - \lambda b_i^n(k)} > \frac{s_i^n(y_i)}{s_i^n(k)}$, $\forall k \neq y_i$. On the other hand, the inequality $\frac{s_i^l(y_i) - \lambda b_i^l(y_i)}{s_i^l(k) - \lambda b_i^l(k)} > \frac{s_i^l(y_i)}{s_i^l(k)}$, $\forall k \neq y_i$ indicates that the reduction also amplifies the importance of edge weights connecting labeled vertices within the same class while diminishing the importance of those labeled vertices from different classes. To better illustrate the role of variance-enlarged regularization, consider the edge weights that satisfy $w_{ij} = 1.0$ if $y_i = y_j$ and $w_{ij} = 0.1$ if $y_i \neq y_j$, with setting $\lambda q_i q_j = 0.1$. In this scenario, we have $s_i^m(y_i) - \lambda b_i^m(y_i) = 0.9$ and $s_i^m(y_i) - \lambda b_i^m(y_i) = 0$, $\forall m \in \{n, l\}, k \neq y_i$. This suggests that the solutions will be a weighted combination of vertices within the same class and eliminate the negative contributions from other classes, especially in the presence of a large number of unlabeled data.

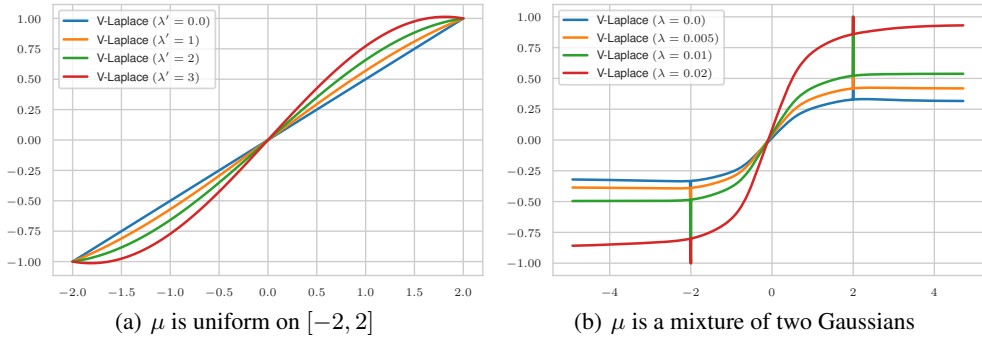

(a) $\mu$ is uniform on $[-2, 2]$       (b) $\mu$ is a mixture of two Gaussians

Figure 1: Behavior of the V-Laplace solutions with different $\lambda$ values under a change of input density $\mu(x)$, where two labeled points at $-2$ and $2$ are given with label $-1$ and $1$, respectively. (a) The solutions of a 1-dimension problem with $\mu$ being uniform on $U[-2,2]$. (b) The solutions of a 1-dimension problem with $\mu$ being a mixture of two 1-dimension Gaussians $\mathcal{N}(-2,1)$ and $\mathcal{N}(2,1)$.

## 4.2 THE VARIATIONAL CASE

We further use techniques and ideas from PDE theory to yield deeper understanding of the behavior of variance-enlarged Poisson learning in the context of a more general objective incorporating the $\ell_p$ distance ($p \geq 2$) (Oberman, 2013).

Our analysis focused on the formulations in the asymptotic limit as the graph size goes to infinity (*i.e.*, $n \to \infty$) and the bandwidth of the edge weight $w_{ij} = \varphi\left(\frac{\|x_i - x_i\|_2}{h}\right)$ goes to zero (*i.e.*, $h \to 0$) (Nadler et al., 2009; El Alaoui et al., 2016). In particular, we consider the problem

$$\lim_{h \to 0} \lim_{n \to \infty} \tfrac{1}{n^2 h^{p+d} C_p} \mathcal{J}_p(u) - \lambda \cdot \mathrm{Var}[u], \tag{4.2}$$

where $\mathcal{J}_p(u) = \sum_{ij} w_{ij}^p \|u(x_i) - u(x_j)\|_p^p$, $\mathrm{Var}[u] = \sum_{i=1}^n q_i \|u(x_i) - \sum_{j=1}^n q_j u(x_j)\|^2$, and $C_p = \frac{1}{d^{p/2}} \int \|z\|_2^p \varphi(\|z\|_2)^p \mathrm{d}z$ is a constant.

When $n \to \infty$ and $h \to 0$, follow the asymptotic behavior of the objective $\mathcal{J}_p(u)$ in (Nadler et al., 2009; El Alaoui et al., 2016), we have $\lim_{h \to 0} \lim_{n \to \infty} \frac{1}{n^2 h^{p+d} C_p} \mathcal{J}_p(u) = \int \|\nabla u(x)\|_{2,p}^p \mu^2(x) \mathrm{d}x$ for even $p$ (please see Theorem 1 in (El Alaoui et al., 2016)). Thus, the functional with respect to $u(x)$ is formulated as

$$\mathcal{I}_p(u) = \int \|\nabla u(x)\|_{2,p}^p \mu^2(x) \mathrm{d}x - \frac{\lambda}{2}\left[\int \|u(x)\|_2^2 q(x) \mathrm{d}x - \left\|\int u(x) q(x) \mathrm{d}x\right\|_2^2\right], \tag{4.3}$$

where the variance term is described as $\lim_{h \to 0} \lim_{n \to \infty} \mathrm{Var}[u] = \int \|u(x)\|_2^2 q(x) \mathrm{d}x - \|\int u(x) q(x) \mathrm{d}x\|_2^2$, and $q(x)$ is the weight assigned to $x$ for calculating the variance.

Additionally, $q(x)$ is somewhat restricted to ensure that $q(x)$ together with a sufficiently small $\lambda$ can guarantee the convexity of Eq. 4.3. Our first result characterizes the range of $\lambda$ as follows:

**Theorem 4.4.** *For $p \geq 2$, there exists a constant $C$ depending on $\Omega$, $q(x)$ and $d$, such that, when $\lambda \leq \inf_{x \in \Omega, u \in \mathcal{F}} \frac{p \cdot \|\nabla u(x)\|_{2,p}^{p-2} \mu^2(x)}{C^2 q(x)}$, the functional in Eq. 4.3 shows convexity with respect to $u$.*

As can be seen, when $p = 2$, the range can be simplified as $\lambda \leq \inf_{x \in \Omega} \frac{2\mu^2(x)}{C^2 q(x)}$, which depends on $q(x)$ and $\mu(x)$. Our next main result characterizes the solutions of the optimization problem $\inf_u \mathcal{I}(u)$ in terms of a PDE referred to as the (weighted) $p$-Poisson equation.

**Theorem 4.5.** *Suppose the density $\mu$ is bounded and continuously differentiable. If $\lambda \leq \inf_{x \in \Omega, u \in \mathcal{F}} \frac{p \cdot \|\nabla u(x)\|_{2,p}^{p-2} \mu^2(x)}{C^2 q(x)}$, then any twice-differentiable minimizer $u : \mathbb{R}^d \to \mathbb{R}^k$ of the functional in Eq. 4.3 must satisfy the Euler-Lagrange equation*

$$\mathrm{div}\left(\mu^2(x) D_p(\nabla u) \nabla u(x)\right) = \frac{\lambda q(x)}{p}\left[\int q(x) u(x) \mathrm{d}x - u(x)\right], \tag{4.4}$$

Table 1: Classification accuracy (%) of different methods on MNIST with several labels per class ($\{1, 2, 3, 4, 5\}$). The results described as "mean (std)" are run over 100 trials. Results achieved by V-Laplace and V-Poisson that outperform their original ones are **boldfaced** and the the best results are underlined.

| # Labels per class | 1 | 2 | 3 | 4 | 5 |
|---|---|---|---|---|---|
| Laplace/LP (Zhu et al., 2003) | 16.7 (8.0) | 30.8 (12.8) | 47.0 (15.0) | 64.9 (13.3) | 76.2 (10.3) |
| Nearest Neighbor | 56.2 (4.9) | 66.3 (3.8) | 70.2 (2.6) | 73.4 (2.7) | 75.5 (2.2) |
| Random Walk (Zhou & Schölkopf, 2004) | 85.0 (4.6) | 91.0 (2.5) | 92.7 (1.6) | 93.9 (1.1) | 94.5 (0.8) |
| MBO (Garcia-Cardona et al., 2014) | 19.4 (6.2) | 29.3 (6.9) | 40.2 (7.4) | 50.7 (6.0) | 59.2 (6.0) |
| WNLL (Shi et al., 2017) | 59.2 (14.3) | 87.8 (5.8) | 93.5 (2.6) | 95.4 (1.2) | 95.8 (0.7) |
| Centered Kernel (Mai & Couillet, 2018) | 84.3 (5.0) | 90.8 (2.5) | 92.5 (1.8) | 93.9 (1.2) | 94.7 (0.9) |
| Sparse LP (Jung et al., 2016) | 14.0 (5.5) | 14.0 (4.0) | 14.5 (4.0) | 18.0 (5.9) | 16.2 (4.2) |
| p-Laplace (Rios et al., 2019) | 72.7 (4.8) | 82.7 (3.4) | 86.1 (2.3) | 88.3 (1.7) | 89.7 (1.3) |
| Poisson (Calder et al., 2020) | 91.3 (4.2) | 94.3 (1.7) | 95.0 (1.1) | 95.5 (0.7) | 95.8 (0.6) |
| **V-Laplace** | **92.0 (4.3)** | **95.3 (1.5)** | **95.9 (0.8)** | **96.2 (0.5)** | **96.3 (0.5)** |
| **V-Poisson** | **93.6 (3.9)** | **95.7 (1.5)** | **96.1 (0.6)** | **96.4 (0.4)** | **96.4 (0.4)** |

Table 2: Classification accuracy (%) of different methods on FashionMNIST with several labels per class ($\{1, 2, 3, 4, 5\}$). The results described as "mean (std)" are run over 100 trials. Results achieved by V-Laplace and V-Poisson that outperform their original ones are **boldfaced** and the the best results are underlined.

| # Labels per class | 1 | 2 | 3 | 4 | 5 |
|---|---|---|---|---|---|
| Laplace/LP (Zhu et al., 2003) | 17.0 (6.6) | 31.7 (10.0) | 43.3 (8.4) | 52.8 (6.9) | 59.3 (5.7) |
| Nearest Neighbor | 43.9 (4.3) | 49.6 (3.3) | 52.7 (3.0) | 55.0 (2.4) | 56.9 (2.7) |
| Random Walk (Zhou & Schölkopf, 2004) | 57.1 (4.8) | 63.1 (4.0) | 66.3 (2.8) | 68.5 (2.5) | 70.1 (2.2) |
| MBO (Garcia-Cardona et al., 2014) | 15.7 (4.1) | 20.1 (4.6) | 25.7 (4.9) | 30.7 (4.9) | 34.8 (4.3) |
| WNLL (Shi et al., 2017) | 43.0 (7.6) | 58.6 (5.1) | 64.0 (3.4) | 67.1 (3.4) | 69.6 (2.7) |
| Centered Kernel (Mai & Couillet, 2018) | 36.6 (4.2) | 47.2 (4.4) | 53.5 (3.9) | 58.4 (3.3) | 61.6 (3.4) |
| Sparse LP (Jung et al., 2016) | 14.0 (5.5) | 14.0 (4.0) | 14.5 (4.0) | 18.0 (5.9) | 16.2 (4.2) |
| p-Laplace (Rios et al., 2019) | 52.1 (4.8) | 58.4 (3.7) | 62.0 (3.0) | 64.3 (2.5) | 66.0 (2.5) |
| Poisson (Calder et al., 2020) | 60.4 (4.7) | 66.3 (4.0) | 68.9 (2.7) | 70.7 (2.4) | 72.2 (2.2) |
| **V-Laplace** | **60.6 (5.0)** | **66.3 (4.2)** | **69.2 (2.8)** | **71.0 (2.8)** | **72.6 (2.3)** |
| **V-Poisson** | **61.3 (4.9)** | **67.1 (4.2)** | **69.7 (2.8)** | **71.3 (2.7)** | **72.9 (2.3)** |

*where $\mathrm{div}(\nabla f) = (\sum_{i=1}^{d} \partial_{x_i}^2 f_1, \sum_{i=1}^{d} \partial_{x_i}^2 f_2, ..., \sum_{i=1}^{d} \partial_{x_i}^2 f_k)^\top$ for $f : \mathbb{R}^d \to \mathbb{R}^k$ and $D_p(\nabla u) = \mathrm{diag}\left(\|\nabla u_1(x)\|_2^{p-2}, \|\nabla u_2(x)\|_2^{p-2}, ..., \|\nabla u_k(x)\|_2^{p-2}\right)$. Furthermore, if the distribution $\mu$ has full support, then this equation can be solved equivalently by addressing*

$$\mathrm{div}(\nabla u(x)) + 2\nabla u(x)\nabla \log \mu(x) + (p-2)\Delta_\infty u(x) + \frac{\lambda q(x)}{p\mu^2(x)} \cdot D_p(\nabla u)^{-1} u(x) = 0, \quad (4.5)$$

*and $\int u(x)q(x)dx = 0$, where $\Delta_\infty u = \left(\frac{\nabla u_1^\top \nabla^2 u_1 \nabla u_1}{\|\nabla u_1\|_2^2}, \frac{\nabla u_2^\top \nabla^2 u_2 \nabla u_2}{\|\nabla u_2\|_2^2}, ..., \frac{\nabla u_k^\top \nabla^2 u_k \nabla u_k}{\|\nabla u_k\|_2^2}\right).$*

**Remark.** Here, we specialize to the case $d = k = 1$ and $p = 2$ to better show the behavior of enlarging variance, then $\Delta_\infty u = \mathrm{div}(\nabla u) = \frac{\mathrm{d}^2 u}{\mathrm{d}x^2}$, and Eq. 4.5 reduces to the second order ODE: $2\mu^2(x)u''(x) + 4\mu(x)\mu'(x)u'(x) + \lambda q(x)u(x) = 0$. If we further let $\mu$ be the uniform distribution and $q(x) = \mu(x)$, we have $u''(x) + \lambda u(x) = 0$ [2]. As illustrated in Fig. 1(a), the solution of Laplace equation ($\lambda' = 0$) is a straight line which is the shortest distance between two labeled points at boundary. This behavior arises from the fact that the solutions of Laplace equation are harmonic functions that have no local maxima or minima. The existence of variance-enlarged regularization change this point, while it makes the confidence of prediction becomes higher as $\lambda$ increases. Particularly noteworthy in Fig. 1(b), when $\mu$ is a mixture of two Gaussian, VPL mitigates the occurrence of spikes of labeled points and non-informative predictions for unlabeled points. This partially explains the effectiveness of enlarging variance for low label rate problems.

## 5 Experiments

In this section, we provide extensive experiments to validate the effectiveness of variance-enlarged Poisson learning with extremely sparse labeled data (where the number of label per class is taken in

---

[2]We can easily derive that $u(x) = C_1 \cos \sqrt{\lambda'} x + C_2 \sin \sqrt{\lambda'} x$.

Table 3: Classification accuracy (%) of different methods on CIFAR-10 with several labels per class ($\{1, 2, 3, 4, 5\}$). The results described as "mean (std)" are run over 100 trials. Results achieved by V-Laplace and V-Poisson that outperform their original ones are **boldfaced** and the the best results are underlined.

| # LABELS PER CLASS | 1 | 2 | 3 | 4 | 5 |
|---|---|---|---|---|---|
| LAPLACE/LP (ZHU ET AL., 2003) | 10.3 (1.2) | 10.8 (1.7) | 11.8 (2.7) | 13.0 (4.0) | 13.1 (3.4) |
| NEAREST NEIGHBOR | 29.4 (3.9) | 33.4 (3.3) | 35.1 (2.9) | 36.4 (2.3) | 37.4 (2.4) |
| RANDOM WALK (ZHOU & SCHÖLKOPF, 2004) | 37.5 (5.1) | 44.6 (3.5) | 48.4 (3.7) | 51.1 (3.0) | 52.8 (2.8) |
| MBO (GARCIA-CARDONA ET AL., 2014) | 14.2 (4.1) | 19.3 (5.2) | 24.3 (5.6) | 28.5 (5.6) | 33.5 (5.7) |
| WNLL (SHI ET AL., 2017) | 14.9 (4.8) | 24.9 (6.9) | 33.2 (6.7) | 38.4 (7.0) | 42.4 (5.5) |
| CENTERED KERNEL (MAI & COUILLET, 2018) | 35.6 (5.4) | 42.7 (5.4) | 46.0 (3.6) | 48.6 (3.2) | 50.1 (2.7) |
| SPARSE LP (JUNG ET AL., 2016) | 11.8 (2.4) | 12.3 (2.4) | 11.1 (3.3) | 14.4 (3.5) | 11.0 (2.9) |
| P-LAPLACE (RIOS ET AL., 2019) | 34.7 (4.7) | 41.3 (3.5) | 44.6 (3.6) | 47.2 (2.8) | 48.8 (2.8) |
| POISSON (CALDER ET AL., 2020) | 39.1 (5.4) | 45.4 (3.9) | 48.5 (3.6) | 51.2 (3.0) | 52.9 (2.8) |
| **V-LAPLACE** | **33.9 (5.6)** | **40.5 (4.3)** | **44.0 (4.9)** | **46.6 (4.8)** | **47.8 (4.3)** |
| **V-POISSON** | **41.4 (5.4)** | **48.5 (4.1)** | **51.7 (3.7)** | **54.7 (3.1)** | **56.3 (2.8)** |

Table 4: Classification accuracy with different label rates on Cora and CiteSeer. The best resutls are **boldfaced**.

| #DATASET | # LABELS PER CLASS | 1 | 2 | 3 | 4 | 5 |
|---|---|---|---|---|---|---|
| CORA | GCN (KIPF & WELLING, 2017) | 51.1 (5.9) | 61.1 (3.6) | 65.6 (5.5) | 69.2 (1.5) | 71.3 (3.2) |
| | GAT (VELIČKOVIĆ ET AL., 2018) | 50.5 (3.4) | 59.7 (4.1) | 63.1 (6.8) | 66.7 (1.9) | 68.8 (3.7) |
| | GPN (WAN ET AL., 2021) | 53.2 (8.0) | 61.3 (4.3) | 63.5 (7.2) | 64.1 (5.3) | 65.8 (6.5) |
| | **V-GPN** | **58.9 (7.3)** | **64.1 (5.0)** | **67.6 (5.0)** | **70.5 (2.4)** | **72.8 (1.7)** |
| CITESEER | GCN (KIPF & WELLING, 2017) | 43.8 (6.1) | 51.4 (4.5) | 56.2 (5.7) | 59.5 (4.9) | 61.2 (2.2) |
| | GAT (VELIČKOVIĆ ET AL., 2018) | 42.0 (3.6) | 51.1 (4.6) | 53.6 (1.5) | 58.7 (5.0) | 58.5 (3.9) |
| | GPN (WAN ET AL., 2021) | 39.8 (9.0) | 46.6 (9.5) | 47.4 (6.4) | 46.3 (5.0) | 48.9 (4.3) |
| | **V-GPN** | **49.5 (4.4)** | **55.4 (5.6)** | **57.6 (2.2)** | **62.1 (2.0)** | **63.2 (2.4)** |

$\{1, 2, 3, 4, 5\}$) on two tasks: graph-based semi-supervised learning on MNIST (LeCun et al., 1998), FashionMNIST (Xiao et al., 2017), and CIFAR-10 (Krizhevsky & Hinton, 2009), and semi-supervised node classification with GNNs on the citation network datasets—Cora and CiteSeer (Sen et al., 2008). **More experimental results and details can be found in the Appendix B.**

**Results on Graph-based Semi-supervised Learning.** Tables 1, 2 and 3 report the average accuracy and deviation while randomly selecting extremely sparse labeled data points on MNIST, Fashion-MNIST, and CIFAR-10, respectively. These results show that V-Laplace learning and V-Poisson learning have achieved very prominent results compared with Laplace learning and Poisson learning, which demonstrates the effectiveness of VPL. Notably, V-Poisson consistently outperforms other methods by an obvious margin across all cases. We also provide results on SimSiam (Chen & He, 2021) pretrained features on CIFAR-10, where the details can be found in Appendix B.1.

**Results on Semi-supervised Node Classification with GNNs.** The experimental results on Cora and CiteSeer are shown in Table 4. As can be seen, our proposed V-GPN achieves substantial performance gains at different label rates when compared with the baselines GCN (Kipf & Welling, 2017), GAT (Veličković et al., 2018) and GPN (Wan et al., 2021). In particular, the margin between our proposed framework and the best baseline method can exceed 5% on Cora and CiteSeer datasets given one labeled node per class, which demonstrates that V-GPNs could effectively enhance the learning performance of GNNs with extremely sparse labeled data.

## 6 CONCLUSION

In this study, to address the challenges associated with sparse labeled data in graph-based semi-supervised learning, we introduced Variance-enlarged Poisson Learning (VPL) to enlarge the variance of labels, which intuitively increases the dispersion of labels away from their average value. Based on this variance-enlarged framework, we present two efficient algorithms, namely V-Laplace and V-Poisson, which are tailored for improving Laplace learning and Poisson learning, respectively. Additionally, we extend VPL to enhance semi-supervised node classification with graph neural networks, introducing Variation-enlarged Graph Poisson Networks. Furthermore, we conduct a theoretical exploration of VPL in both finite and infinite data cases, yielding deeper insights into understanding the behavior of VPL. Through extensive experiments, we demonstrate the effectiveness and superior performance of VPL in scenarios characterized by extremely sparse labeled data.

## ACKNOWLEDGE

This work was supported in part by National Natural Science Foundation of China under Grants 92270116, 62071155 and 623B2031, and in part by the Fundamental Research Funds for the Central Universities (Grant No.HIT.OCEF.2023043 and HIT.DZJJ.2023075).

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

# Appendix for "Variance-enlarged Poisson Learning"

## A    PROOFS OF LEMMAS AND THEOREMS

### A.1    PROOF OF THEOREM 3.1

**Theorem 3.1** *If $\lambda \leq \min_{ij} \frac{w_{ij}}{q_i q_j}$, the graph $G$ with the weight matrix $W' = W - \lambda q q^\top$ (where $q = (q_1, ..., q_n)^\top$) is connected and the Markov chain induced by the random walk is aperiodic, we have: (1) For the iterations $u_{T+1}(x_i) = u_T(x_i) + d_i^{-1}(\lambda q_i(u_T(x_i) - \overline{u}_T) - \mathcal{L}u_T(x_i))$ of V-Laplace learning in Algorithm 1, $u_T$ converges to the solution of the problem in Eq. 3.3; (2) For the iterations $u_{T+1}(x_i) = u_T(x_i) + d_i^{-1}(\sum_{j=1}^l (y_j - \overline{y})\mathbb{I}(i = j) + \lambda q_i(u_T(x_i) - \overline{u}_T) - \mathcal{L}u_T(x_i))$ of V-Poisson learning in Algorithm 2, then $u_T \to u$ as $T \to \infty$, where $u$ is the solution of Poisson equation 3.4 satisfying $\sum_{i=1}^n d_i u(x_i) = 0$.*

*Proof.* In this proof, we first perform an analysis of the quadratic property of V-Laplace and then show the convergence in Algorithm 1 that conforms to the gradient descent algorithm. For the convergence of V-poisson, our proof depends on the characterization of the iterations, such as the identity $\sum_{i=1}^n d_i u_T(x_i) = 0$.

(1) For the V-Laplace learning, the optimization problem can be formulated as minimizing $\sum_{i=1}^n \sum_{j=1}^n (w_{ij} - \lambda q_i q_j)\|u(x_i) - u(x_j)\|_2^2$ in accordance with Lemma 4.1. Since the iterations in Algorithm 1 follows the GD sequence with the learning rate $\eta = \frac{1}{d_i} < \frac{1}{d_i - \lambda q_i}$, we know that Algorithm 1 will converge to the condition $\mathcal{L}u(x_i) = \lambda q_i(u(x_i) - \overline{u})$ for $l + 1 \leq i \leq n$.

(2) For the V-Poisson learning, the iteration in Algorithm 2 shows that

$$d_i(u_{T+1}(x_i) - u_T(x_i)) + \mathcal{L}u_T(x_i) = \lambda q_i(u_T(x_i) - \overline{u}_T) + \sum_{j=1}^m (y_j - \overline{y})\mathbb{I}(i = j). \tag{A.1}$$

Summing both sides over $i = 1, ..., n$, we have

$$\sum_{i=1}^n d_i u_{T+1}(x_i) = \sum_{i=1}^n d_i u_T(x_i) = ... = \sum_{i=1}^n d_i u_0(x_i) = 0, \tag{A.2}$$

because $u_0(x_i)$ is initialized as zero in Algorithm 2.

Let $u$ be the solution of the Poisson equation

$$\mathcal{L}u(x_i) = \sum_{j=1}^l (u_j - \overline{y})\mathbb{I}(i = j) + \lambda q_i(u(x_i) - \overline{u}) \tag{A.3}$$

satisfying $\sum_{i=1}^n d_i u(x_i) = 0$. Define $v_T(x_i) = (d - \lambda q_i)(u_T(x_i) - u(x_i))$, combining Eq. A.1 and Eq. A.3, we obtain

$$v_T(x_i) = \sum_{j=1}^n \frac{w_{ij} - \lambda q_i q_j}{d_j - \lambda q_j} v_{T-1}(x_j), \tag{A.4}$$

and $\sum_{j=1}^n v_T(x_j) = 0$. Since the graph $G$ with the weight matrix $W' = W - \lambda q q^\top$ is connected and the Markov chain induced by the corresponding random walk is aperiodic, we have $\lim_{T \to} v_T(x_i) = \frac{d_i - \lambda q_i}{\sum_{j=1}^n d_j - \lambda} \sum_{j=1}^n v_0(x_j) = 0$, *i.e.*, $u_T \to u$ as $T \to \infty$, which completes the proof.d

$\square$

### A.2    PROOF OF LEMMA 4.1

**Lemma 4.1.** *The solution of the Poisson equation in Eq. 3.2 is equivalent to the minimizer of the objective $\sum_{i=1}^n \sum_{j=1}^n (w_{ij} - \lambda q_i q_j)\|u(x_i) - u(x_j)\|_2^2$.*

*Proof.* When $L(u,v) = \|u - v\|_2^2$, we have

$$\frac{1}{2}\sum_{i=1}^{n}\sum_{j=1}^{n}w_{ij}L(u(x_i), u(x_j)) - \lambda \cdot \mathrm{Var}[u]$$

$$=\frac{1}{2}\sum_{i=1}^{n}\sum_{j=1}^{n}w_{ij}\|u(x_i) - u(x_j)\|_2^2 - \lambda\sum_{i=1}^{n}q_i\|u(x_i) - \overline{u}\|_2^2$$

$$=\frac{1}{2}\sum_{i=1}^{n}\sum_{j=1}^{n}w_{ij}\|u(x_i) - u(x_j)\|_2^2 - \lambda\sum_{i=1}^{n}q_i\left\|u(x_i) - \sum_{j=1}^{n}q_ju(x_j)\right\|_2^2 \tag{A.5}$$

$$=\frac{1}{2}\sum_{i=1}^{n}\sum_{j=1}^{n}w_{ij}\|u(x_i) - u(x_j)\|_2^2 - \lambda\left[\sum_{i=1}^{n}q_i\|u(x_i)\|_2^2 - \sum_{i=1}^{n}\sum_{j=1}^{n}q_iq_j\langle u(x_i), u(x_j)\rangle\right]$$

$$=\frac{1}{2}\sum_{i=1}^{n}\sum_{j=1}^{n}(w_{ij} - \lambda q_iq_j)\|u(x_i) - u(x_j)\|_2^2$$

which completes the proof. $\qquad\square$

## A.3 PROOF OF THEOREM 4.4

**Theorem 4.4** *For $p \geq 2$, there exists a constant $C$, depending on $\Omega$ and $d$, such that, when $\lambda \leq \inf_{x\in\Omega, u\in\mathcal{F}}\frac{p\cdot\|\nabla u(x)\|_{2,p}^{p-2}\mu^2(x)}{C^2q(x)}$, the functional in Eq. 4.3 shows convexity with respect to $u$.*

*Proof.* Without any loss of generality, one can assume in Eq. 4.3 $u$ is a scalar function. It is well known that a function $f$ on a finite or infinite dimensional space $\mathcal{F}$ is convex if and only if for any $u, v \in \mathcal{F}$, $f(u + tv)$ is a one dimensional convex function in $t$ (Boyd & Vandenberghe, 2004). So given any $v \in \mathcal{F}$ and small $\epsilon > 0$, define that

$$\mathcal{I}_p(u + \epsilon v) = \int \|\nabla(u + \epsilon v)(x)\|_2^p\mu^2(x)\mathrm{d}x$$
$$- \frac{\lambda}{2}\left[\int \|(u + \epsilon v)(x)\|_2^2q(x)\mathrm{d}x - \left\|\int (u + \epsilon v)(x)q(x)\mathrm{d}x\right\|_2^2\right]. \tag{A.6}$$

We can derive the second-order deviate of $\mathcal{I}$ with respect to $\epsilon$ as follows

$$\mathcal{I}_p''(u + \epsilon v) = p(p - 2)\int \|\nabla(u + \epsilon v)(x)\|_2^{p-4}\left(\langle\nabla u, \nabla v\rangle(x) + \epsilon\|\nabla v(x)\|_2^2\right)^2\mu^2(x)\mathrm{d}x$$
$$+ p\int \|\nabla(u + \epsilon v)(x)\|_2^{p-2}\|\nabla v(x)\|_2^2\mu^2(x)\mathrm{d}x \tag{A.7}$$
$$- \lambda\int v^2(x)q(x)\mathrm{d}x + (\lambda\int v(x)q(x)\mathrm{d}x)^2$$

As aforementioned, $\mathcal{I}$ is a convex function on $u$ if and only if $\mathcal{I}''(u + \epsilon v) \geq 0$ for any $v$, leading us to investigate

$$\mathcal{I}''(u + \epsilon v) \geq p\int \|\nabla(u + \epsilon v)(x)\|_2^{p-2}\|\nabla v(x)\|_2^2\mu^2(x)\mathrm{d}x$$
$$- \lambda\int v^2(x)q(x)\mathrm{d}x + (\lambda\int v(x)q(x)\mathrm{d}x)^2 \tag{A.8}$$

On one hand, by Poincaré–Wirtinger inequality on the weighted Sobolev space $W_0^{1,p}(\Omega)$ satisfying $\left(\int \|\nabla f(x)\|_2^pq(x)\mathrm{d}x\right)^{1/p} < \infty$, there exists a constant $C > 0$ depending on $\Omega$, $d$ and $q$, such that

$$\left[\int \left(v(x) - \int v(x)q(x)\mathrm{d}x\right)^2q(x)\mathrm{d}x\right]^{1/2} \leq C\left(\int \|\nabla v(x)\|_2^2q(x)\mathrm{d}x\right)^{1/2}. \tag{A.9}$$

On the other hand, by mean value theorem, there exists $\xi$, such that

$$\int \|\nabla(u + \epsilon v)(x)\|_2^{p-2} \|\nabla v(x)\|_2^2 \mu^2(x) \mathrm{d}x = \frac{\|\nabla(u + \epsilon v)(\xi)\|_2^{p-2} \mu^2(\xi)}{q(\xi)} \int \|\nabla v(x)\|_2^2 q(x) \mathrm{d}x. \tag{A.10}$$

Together with Eq. A.9 and Eq. A.10, it immediately implies that when $\lambda \leq \inf_{x \in \Omega, u \in \mathcal{F}} \frac{p \cdot \|\nabla u(x)\|_2^{p-2} \mu^2(x)}{C^2 q(x)} \leq \frac{p \cdot \|\nabla(u + \epsilon v)(\xi)\|_2^{p-2} \mu^2(\xi)}{C^2 q(\xi)}$, $J$ is a convex function on $u$. $\qquad\square$

## A.4  PROOF OF THEOREM 4.5

**Theorem 4.5.**    *Suppose the density $\mu$ is bounded and continuously differentiable. If $\lambda \leq \inf_{x \in \Omega, u \in \mathcal{F}} \frac{p \cdot \|\nabla u(x)\|_{2,p}^{p-2} \mu^2(x)}{C^2 q(x)}$, then any twice-differentiable minimizer $u : \mathbb{R}^d \to \mathbb{R}^k$ of the functional in Eq. 4.3 must satisfy the Euler-Lagrange equation*

$$p \cdot \mathrm{div}\left(\mu^2(x) D_p(\nabla u)\nabla u(x)\right) + \lambda u(x)q(x) - \lambda q(x)\int q(x)u(x)dx = 0. \tag{A.11}$$

*where $\mathrm{div}(\nabla f) = (\sum_{i=1}^d \partial_{x_i}^2 f_1, \sum_{i=1}^d \partial_{x_i}^2 f_2, ..., \sum_{i=1}^d \partial_{x_i}^2 f_k)^\top$ for $f : \mathbb{R}^d \to \mathbb{R}^k$ and $D_p(\nabla u) = \mathrm{diag}\left(\|\nabla u_1(x)\|_2^{p-2}, \|\nabla u_2(x)\|_2^{p-2}, ..., \|\nabla u_k(x)\|_2^{p-2}\right)$. If moreover, the distribution $\mu$ has full support, then this equation is equivalent to*

$$\mathrm{div}(\nabla u(x)) + 2\nabla u(x)\nabla \log \mu(x) + (p-2)\Delta_\infty u(x) + \frac{\lambda q(x)}{p\mu^2(x)} \cdot D_p(\nabla u)^{-1} u(x) = 0, \tag{A.12}$$

*and $\int u(x)q(x)dx = 0$, where $\Delta_\infty u = \left(\frac{\nabla u_1^\top \nabla^2 u_1 \nabla u_1}{\|\nabla u_1\|_2^2}, \frac{\nabla u_2^\top \nabla^2 u_2 \nabla u_2}{\|\nabla u_2\|_2^2}, ..., \frac{\nabla u_k^\top \nabla^2 u_k \nabla u_k}{\|\nabla u_k\|_2^2}\right)$.*

*Proof.* Without any loss of generality, one can assume in Eq. 4.3 $u$ is a scalar function, since we can analyze each component of $u$ independently. Let $I_p(u) = \int \|\nabla u(x)\|_2^p \mu^2(x)\mathrm{d}x - \frac{\lambda}{2}\left[\int (u(x))^2 q(x)\mathrm{d}x - \left(\int u(x)q(x)\mathrm{d}x\right)_2^2\right]$. The function $u$ is a minimizer of the functional $I_p$ if for all test functions $h$ and all-sufficient small real numbers $\epsilon > 0$, we have $I_p(u + \epsilon h) \geq I_p(u)$. Moreover, by a Taylor series expansion, we have

$$\begin{aligned}\mathcal{I}_p(u + \epsilon h) = &\mathcal{I}_p(u) + p\epsilon \int \langle \nabla u(x), \nabla h(x)\rangle \cdot \|\nabla u(x)\|_2^{p-2}\mu^2(x)\mathrm{d}x \\ &- \lambda\epsilon \left(\int u(x)h(x)q(x)\mathrm{d}x - \int u(x)q(x)\mathrm{d}x \int h(x)q(x)\mathrm{d}x\right) + \mathcal{O}(\epsilon^2),\end{aligned} \tag{A.13}$$

where the $\mathcal{O}(\epsilon^2)$ term is non-negative by convexity of $I_p$. Hence, the function $u$ is a minimizer if and only if

$$\begin{aligned}0 = &p\int \langle \nabla u(x), \nabla h(x)\rangle \cdot \|\nabla u(x)\|_2^{p-2}\mu^2(x)\mathrm{d}x - \lambda \int u(x)h(x)q(x)\mathrm{d}x \\ &+ \lambda \int u(x)q(x)\mathrm{d}x \int h(x)q(x)\mathrm{d}x,\end{aligned} \tag{A.14}$$

for all testing functions $h$. By integrating by parts and choosing $h$ to vanish on the boundary of $\Omega$, the integration by parts formula shows that the above quantity is equal to

$$\int \langle \nabla u(x), \nabla h(x)\rangle \cdot \|\nabla u(x)\|_2^{p-2}\mu^2(x)\mathrm{d}x = -\int \mathrm{div}\left(\mu^2(x)\|\nabla u(x)\|_2^{p-2}\nabla u(x)\right) h(x)\mathrm{d}x, \tag{A.15}$$

where $\mathrm{div}(\nabla f) = \sum_{i=1}^d \partial_{x_i}^2 f$ for $f : \mathbb{R}^d \to \mathbb{R}$.

This expression has to vanish for all test functions $h$ (that vanish on the boundary), which implies the Euler-Lagrange equation

$$p \cdot \mathrm{div}(\mu^2(x)\|\nabla u(x)\|_2^{p-2}\nabla u(x)) + \lambda u(x)q(x) - \lambda q(x)\int u(x)q(x)\mathrm{d}x = 0. \tag{A.16}$$

We now further manipulate this equation so as to obtain the $p$-Poisson equation. In particular, some straightforward computations yield

$$\partial_{x_i}(\mu^2 \|\nabla f\|_2^{p-2}\partial_{x_i}f)(x) = \partial_{x_i}(\mu^2(x)\|\nabla u(x)\|_2^{p-2})\partial_{x_i}u(x) + \mu^2(x)\|\nabla u(x)\|_2^{p-2}\partial_{x_i}^2 u(x), \tag{A.17}$$

and

$$\partial_{x_i}(\mu^2\|\nabla f\|_2^{p-2})(x) = 2\partial_{x_i}\mu(x)\cdot\mu(x)\|\nabla u(x)\|^{p-2}$$
$$+ \mu^2(x)(p-2)\left(\sum_{j=1}^{d}\partial_{x_i,x_j}f\partial_{x_j}f\right)\cdot\|\nabla u(x)\|_2^{p-4}. \tag{A.18}$$

Now summing these terms yield

$$\operatorname{div}\left(\mu^2(x)\|\nabla u(x)\|_2^{p-2}\nabla u(x)\right)$$
$$= 2\mu(x)\|\nabla u(x)\|_2^{p-2}\langle\nabla\mu(x),\nabla u(x)\rangle + \mu^2(x)\|\nabla u(x)\|_2^{p-2}\Delta_2 u(x)$$
$$+ (p-2)\mu^2(x)\|\nabla u(x)\|_2^{p-4}\left(\sum_{i,j=1}^{d}\partial_{x_i}f\cdot\partial_{x_i,x_j}f\cdot\partial_{x_j}f\right)(x)$$
$$= \mu^2(x)\|\nabla u(x)\|_2^{p-2}\cdot\left(\Delta_2 u(x) + \frac{2}{\mu(x)}\langle\nabla\mu(x),\nabla u(x)\rangle + (p-2)\Delta_\infty u(x)\right)$$

where $\Delta_2 f := \operatorname{Tr}(\nabla^2 f)$ is the usual 2-Laplacian operator, while $\Delta_\infty f := \frac{\langle\nabla f,\nabla^2 f\nabla f\rangle}{\langle\nabla f,\nabla f\rangle}$ is the $\infty$-Laplacian operator, which is defined to be zero when $\nabla f = 0$.

From the derivation above, let $v(x) = u(x) - \int u(x)q(x)\mathrm{d}x$, we have $\operatorname{div}\left(\mu^2(x)\|\nabla v(x)\|_2^{p-2}\nabla v(x)\right) = \operatorname{div}\left(\mu^2(x)\|\nabla u(x)\|_2^{p-2}\nabla u(x)\right)$, then the Euler-Lagrange equation in Eq. A.16 is equivalent to

$$p\cdot\mu^2(x)\|\nabla v(x)\|_2^{p-2}\cdot\left(\Delta_2 v(x) + \frac{2\langle\nabla\mu(x),\nabla v(x)\rangle}{\mu(x)} + (p-2)\Delta_\infty v(x)\right) + \lambda q(x)v(x) = 0,$$

that is,

$$\Delta_2 v(x) + 2\langle\nabla\log\mu(x),\nabla v(x)\rangle + (p-2)\Delta_\infty v(x) + \frac{\lambda}{p}\cdot\frac{q(x)v(x)}{\mu^2(x)\|\nabla v(x)\|_2^{p-2}} = 0, \tag{A.19}$$

and $\int q(x)v(x)\mathrm{d}x = 0$, which completes the proof.

$\square$

# B  MORE EXPERIMENTAL DETAILS AND RESULTS

## B.1  EXPERIMENTS FOR V-LAPLACE AND V-POISSON

**Datasets.**  We assess V-Laplace and V-Poisson on three datasets: MNIST (LeCun et al., 1998), FashionMNIST (Xiao et al., 2017), and CIFAR-10 (Krizhevsky & Hinton, 2009). We follow the setting in Calder et al. (2020), where variational autoencoders with 3 fully connected layers of sizes (784,400,20) and (784, 400, 30) are used, respectively, followed by a symmetrically defined decoder. The autoencoder is trained for 100 epochs on each dataset. The autoencoder architecture, loss, and training, are similar to (Kingma & Welling, 2013).

For each dataset, we create a graph in the latent feature space using all available data, resulting in $n = 70,000$ nodes for MNIST and FashionMNIST, and $n = 60,000$ nodes for CIFAR-10. The graph is constructed as a $K$-nearest neighbor graph with Gaussian-like weights, calculated as $w_{ij} = \exp(-4\|x_i - x_j\|^2/d_K(x_i)^2) + \epsilon$, where $x_i$ represents the latent variables for example $i$, $d_K(x_i)$ is the distance in the latent space between $x_i$ and its K-th nearest neighbor, and $\epsilon \geq 0$. We set $K = 10$ and $\epsilon = 1$ across all experiments. The weight matrices are symmetrized by replacing $W$ with $W + W^\top$. For simplicity, we directly utilize the off-the-self distance matrices available at

Table 5: MNIST: Average accuracy scores over 100 trials with standard deviation in brackets.

| # LABELS PER CLASS | 1 | 2 | 3 | 4 | 5 |
|---|---|---|---|---|---|
| LAPLACE/LP (ZHU ET AL., 2003) | 16.7 (8.0) | 30.8 (12.8) | 47.0 (15.0) | 64.9 (13.3) | 76.2 (10.3) |
| NEAREST NEIGHBOR | 56.2 (4.9) | 66.3 (3.8) | 70.2 (2.6) | 73.4 (2.7) | 75.5 (2.2) |
| RANDOM WALK (ZHOU & SCHÖLKOPF, 2004) | 85.0 (4.6) | 91.0 (2.5) | 92.7 (1.6) | 93.9 (1.1) | 94.5 (0.8) |
| MBO (GARCIA-CARDONA ET AL., 2014) | 19.4 (6.2) | 29.3 (6.9) | 40.2 (7.4) | 50.7 (6.0) | 59.2 (6.0) |
| WNLL (SHI ET AL., 2017) | 59.2 (14.3) | 87.8 (5.8) | 93.5 (2.6) | 95.4 (1.2) | 95.8 (0.7) |
| CENTERED KERNEL (MAI & COUILLET, 2018) | 84.3 (5.0) | 90.8 (2.5) | 92.5 (1.8) | 93.9 (1.2) | 94.7 (0.9) |
| SPARSE LP (JUNG ET AL., 2016) | 14.0 (5.5) | 14.0 (4.0) | 14.5 (4.0) | 18.0 (5.9) | 16.2 (4.2) |
| P-LAPLACE (RIOS ET AL., 2019) | 72.7 (4.8) | 82.7 (3.4) | 86.1 (2.3) | 88.3 (1.7) | 89.7 (1.3) |
| POISSON (CALDER ET AL., 2020) | 91.3 (4.2) | 94.3 (1.7) | 95.0 (1.1) | 95.5 (0.7) | 95.8 (0.6) |
| **V-LAPLACE** ($\lambda = 0.001$) | **88.9 (4.7)** | 93.8 (1.8) | **94.7 (1.1)** | **95.5 (0.7)** | 95.7 (0.6) |
| **V-LAPLACE** ($\lambda = 0.002$) | **89.1 (4.6)** | 93.9 (1.8) | **94.8 (1.1)** | **95.5 (0.7)** | 95.8 (0.6) |
| **V-LAPLACE** ($\lambda = 0.005$) | **89.7 (4.5)** | 94.1 (1.7) | **94.9 (1.1)** | 95.6 (0.6) | 95.8 (0.6) |
| **V-LAPLACE** ($\lambda = 0.01$) | **90.5 (4.4)** | **94.5 (1.6)** | **95.1 (1.0)** | **95.7 (0.6)** | 95.9 (0.6) |
| **V-LAPLACE** ($\lambda = 0.02$) | **91.3 (4.2)** | **94.9 (1.5)** | **95.4 (0.9)** | **95.9 (0.5)** | **96.1 (0.5)** |
| **V-LAPLACE** ($\lambda = 0.05$) | **91.6 (4.5)** | **95.2 (1.8)** | **95.8 (0.9)** | **96.2 (0.5)** | **96.3 (0.9)** |
| **V-LAPLACE** ($\lambda = 0.1$) | **91.2 (4.9)** | **95.1 (2.3)** | **95.8 (1.1)** | **96.2 (0.7)** | **96.3 (1.0)** |
| **V-LAPLACE** ($\lambda = 0.2$) | **91.4 (4.7)** | **95.2 (2.0)** | **95.8 (1.0)** | **96.3 (0.5)** | **96.3 (1.0)** |
| **V-LAPLACE** ($\lambda = 0.5$) | **92.0 (4.3)** | **95.3 (1.5)** | **95.9 (0.8)** | **96.2 (0.5)** | **96.3 (0.5)** |
| **V-LAPLACE** ($\lambda = 1.0$) | **91.7 (4.0)** | **95.0 (1.4)** | **95.6 (0.9)** | **96.0 (0.6)** | **96.2 (0.5)** |
| **V-POISSON** ($\lambda = 0.001$) | **91.4 (4.1)** | **94.4 (1.6)** | 95.0 (1.1) | 95.6 (0.7) | 95.8 (0.6) |
| **V-POISSON** ($\lambda = 0.002$) | **91.5 (4.1)** | **94.4 (1.6)** | 95.0 (1.0) | 95.6 (0.7) | 95.9 (0.5) |
| **V-POISSON** ($\lambda = 0.005$) | **91.8 (4.1)** | **94.6 (1.6)** | 95.1 (1.0) | **95.7 (0.6)** | 95.9 (0.5) |
| **V-POISSON** ($\lambda = 0.01$) | **92.2 (4.1)** | **94.8 (1.5)** | 95.3 (1.0) | 95.8 (0.6) | 96.0 (0.5) |
| **V-POISSON** ($\lambda = 0.02$) | **92.8 (4.0)** | **95.2 (1.5)** | **95.6 (0.9)** | 96.0 (0.6) | **96.2 (0.5)** |
| **V-POISSON** ($\lambda = 0.05$) | **93.4 (3.9)** | **95.6 (1.4)** | **96.0 (0.7)** | **96.3 (0.5)** | **96.4 (0.4)** |
| **V-POISSON** ($\lambda = 0.1$) | **93.5 (3.9)** | **95.7 (1.5)** | **96.1 (0.7)** | **96.4 (0.4)** | **96.4 (0.5)** |
| **V-POISSON** ($\lambda = 0.2$) | **93.6 (3.9)** | **95.7 (1.5)** | $\overline{96.1 (0.6)}$ | **96.4 (0.4)** | $\overline{96.4 (0.4)}$ |
| **V-POISSON** ($\lambda = 0.5$) | **93.4 (3.9)** | **95.6 (1.4)** | **96.0 (0.7)** | **96.3 (0.5)** | **96.4 (0.4)** |
| **V-POISSON** ($\lambda = 1.0$) | **92.6 (3.8)** | **95.0 (1.4)** | **95.6 (0.9)** | **96.0 (0.7)** | **96.2 (0.6)** |

https://github.com/jwcalder/GraphLearning/tree/master. In term of the total number of iterations, we employ the trick by Calder et al. (2020). Specifically, we add the iteration step $p_{t+1} = \mathbf{W}\mathbf{D}^{-1}p_t$ in the algorithms of V-Laplace and V-Poisson, where $p_0$ is initialized as a vector with ones at the positions of all labeled vertices and zeros elsewhere. The iterations for V-Laplace and V-Poisson are terminated once $\|p_t - p_\infty\|_\infty \leq \frac{1}{n}$ is achieved, just before reaching the mixing time. Herein, $p_\infty = \mathbf{W}\mathbf{1}/(\mathbf{1}^\top \mathbf{W}\mathbf{1})$ represents the invariant distribution.

**Comparisons.** We conduct a comparative evaluation of V-Laplace and V-Poisson against various graph-based semi-supervised learning methodologies, including Laplace learning (Zhu et al., 2003), lazy random walks (Zhou et al., 2003; Zhou & Schölkopf, 2004), multiclass MBO (Garcia-Cardona et al., 2014), weighted nonlocal Laplacian (Shi et al., 2017), Centered Kernel Method (Mai & Couillet, 2018), sparse label propagation (Jung et al., 2016), p-Laplace learning (Rios et al., 2019), and Poisson learning (Calder et al., 2020). For the volume-constrained MBO method, we set the temperature $T = 0.1$. In the case of the Centered Kernel Method, we selected $\alpha$ to be 5% larger than the spectral norm of the centered weight matrix. Throughout our experiments, we ran 100 trials, varying the random seed from 0 to 99. It's worth noting that the results for MBO and Sparse Laplace Propagation are adopted from Calder et al. (2020).

**Hyper-Parameters.** Our proposed V-Laplace and V-Poisson have only one hyper-parameter $\lambda$ to control the strength of enlarging variance. In order to better illustrate its sensitivity, we provide more results in Table 5, Table 6, and Table 7. For the results of V-Laplace in Table 1, Table 2 and Table 3, we set $\lambda = 0.5$, $\lambda = 1.0$ and $\lambda = 0.05$, respectively. For the results of V-Poisson in Table 1 and Table 2, we set $\lambda = 0.2$, $\lambda = 1.0$, respectively. For V-Poisson learning on CIFAR-10, we directly use Poisson learning on the weight matrix which we subtract the elements greater than 0 from their minimum value.

**Experiments on Self-Supervised Learned Representations on CIFAR-10.** Self-supervised models do currently exhibit promising properties in learning representations, and achieving relatively high linear probes. However, when the number of labels is particularly scarce, for instance, with only one label per class, self-supervised models possess good representations but struggle to learn effective classifiers. To illustrate this point, we refer to the open-source code and model parameters associated

Table 6: FashionMNIST: Average accuracy scores over 100 trials with standard deviation in brackets.

| # LABELS PER CLASS | 1 | 2 | 3 | 4 | 5 |
|---|---|---|---|---|---|
| LAPLACE/LP (ZHU ET AL., 2003) | 17.0 (6.6) | 31.7 (10.0) | 43.3 (8.4) | 52.8 (6.9) | 59.3 (5.7) |
| NEAREST NEIGHBOR | 43.9 (4.3) | 49.6 (3.3) | 52.7 (3.0) | 55.0 (2.4) | 56.9 (2.7) |
| RANDOM WALK (ZHOU & SCHÖLKOPF, 2004) | 57.1 (4.8) | 63.1 (4.0) | 66.3 (2.8) | 68.5 (2.5) | 70.1 (2.2) |
| MBO (GARCIA-CARDONA ET AL., 2014) | 15.7 (4.1) | 20.1 (4.6) | 25.7 (4.9) | 30.7 (4.9) | 34.8 (4.3) |
| WNLL (SHI ET AL., 2017) | 43.0 (7.6) | 58.6 (5.1) | 64.0 (3.4) | 67.1 (3.4) | 69.6 (2.7) |
| CENTERED KERNEL (MAI & COUILLET, 2018) | 36.6 (4.2) | 47.2 (4.4) | 53.5 (3.9) | 58.4 (3.3) | 61.6 (3.4) |
| SPARSE LP (JUNG ET AL., 2016) | 14.0 (5.5) | 14.0 (4.0) | 14.5 (4.0) | 18.0 (5.9) | 16.2 (4.2) |
| P-LAPLACE (RIOS ET AL., 2019) | 52.1 (4.8) | 58.4 (3.7) | 62.0 (3.0) | 64.3 (2.5) | 66.0 (2.5) |
| POISSON (CALDER ET AL., 2020) | 60.4 (4.7) | 66.3 (4.0) | 68.9 (2.7) | 70.7 (2.4) | 72.2 (2.2) |
| V-LAPLACE ($\lambda = 0.001$) | **58.8 (5.1)** | **65.1 (4.2)** | **68.3 (2.7)** | **70.2 (2.6)** | **71.8 (2.2)** |
| V-LAPLACE ($\lambda = 0.002$) | **59.0 (5.0)** | **65.2 (4.2)** | **68.4 (2.7)** | **70.2 (2.6)** | **71.9 (2.2)** |
| V-LAPLACE ($\lambda = 0.005$) | **59.3 (5.0)** | **65.4 (4.2)** | **68.5 (2.7)** | **70.4 (2.6)** | **72.0 (2.2)** |
| V-LAPLACE ($\lambda = 0.01$) | **59.7 (5.0)** | **65.7 (4.2)** | **68.7 (2.7)** | **70.5 (2.6)** | **72.1 (2.3)** |
| V-LAPLACE ($\lambda = 0.02$) | **59.7 (5.4)** | **65.9 (4.3)** | **68.8 (3.0)** | **70.6 (2.7)** | **72.3 (2.4)** |
| V-LAPLACE ($\lambda = 0.05$) | **59.2 (5.9)** | **65.6 (4.7)** | **68.3 (3.9)** | **70.1 (3.4)** | **72.1 (2.9)** |
| V-LAPLACE ($\lambda = 0.1$) | **58.9 (6.0)** | **65.2 (4.7)** | **67.9 (4.0)** | **69.6 (3.5)** | **71.7 (3.1)** |
| V-LAPLACE ($\lambda = 0.2$) | **59.2 (5.8)** | **65.3 (4.7)** | **68.0 (3.8)** | **69.7 (3.4)** | **71.7 (3.0)** |
| V-LAPLACE ($\lambda = 0.5$) | **60.1 (5.3)** | **65.9 (4.6)** | **68.7 (3.3)** | **70.4 (3.1)** | **72.2 (2.6)** |
| V-LAPLACE ($\lambda = 1.0$) | **60.6 (5.0)** | **66.3 (4.2)** | **69.2 (2.8)** | **71.0 (2.8)** | **72.6 (2.3)** |
| V-POISSON ($\lambda = 0.001$) | **60.5 (4.7)** | 66.3 (4.0) | 68.9 (2.7) | **70.8 (2.4)** | 72.2 (2.2) |
| V-POISSON ($\lambda = 0.002$) | **60.6 (4.7)** | **66.4 (4.0)** | 68.9 (2.7) | **70.8 (2.4)** | **72.3 (2.2)** |
| V-POISSON ($\lambda = 0.005$) | **60.7 (4.7)** | **66.5 (4.1)** | **69.0 (2.7)** | **70.9 (2.4)** | **72.4 (2.2)** |
| V-POISSON ($\lambda = 0.01$) | **60.9 (4.7)** | **66.7 (4.1)** | **69.2 (2.7)** | **71.0 (2.4)** | **72.5 (2.2)** |
| V-POISSON ($\lambda = 0.02$) | **60.9 (5.0)** | **67.0 (4.2)** | **69.4 (2.8)** | **71.1 (2.6)** | **72.7 (2.3)** |
| V-POISSON ($\lambda = 0.05$) | **60.6 (5.5)** | **66.7 (4.5)** | **69.2 (3.4)** | **70.7 (3.4)** | **72.6 (2.8)** |
| V-POISSON ($\lambda = 0.1$) | **60.5 (5.5)** | **66.4 (4.5)** | **69.0 (3.6)** | 70.4 (3.5) | **72.2 (2.9)** |
| V-POISSON ($\lambda = 0.2$) | **60.7 (5.4)** | **66.5 (4.4)** | **69.1 (3.4)** | 70.5 (3.3) | **72.2 (2.7)** |
| V-POISSON ($\lambda = 0.5$) | **61.2 (5.1)** | **67.0 (4.3)** | **69.5 (3.0)** | **71.0 (2.9)** | **72.7 (2.5)** |
| V-POISSON ($\lambda = 1.0$) | **61.3 (4.9)** | **67.1 (4.2)** | **69.7 (2.8)** | **71.3 (2.7)** | **72.9 (2.3)** |

with an exemplary linear probe achievement of 91.9% on CIFAR-10. The source code and model parameters are available at https://github.com/Reza-Safdari/SimSiam-91.9-top1-acc-on-CIFAR10, which utilizes the SimSiam framework (Chen & He, 2021). Building upon the pretrained model, we conduct two experiments to underscore the effectiveness of graph-based semi-supervised learning in scenarios of label scarcity: (i) We utilize the SimSiam pretrained model to extract features from the entire CIFAR-10 dataset and construct a 10-nearest-neighbors graph on these representations. We then randomly choosing which data points are labeled, and predict unlabeled data for all data points using different graph-based semi-supervised learning approaches; (ii) For comparison, we directly use several randomly chosen labeled data to obtain a linear classifier, and finally use this classifier to evaluate the classification accuracy of all samples.

The experimental results are summarized in Table 8. It is evident that the linear classifier, trained solely on labeled data, outperforms Laplace in scenarios with extremely sparse data. However, it is crucial to note that linear classification exhibits significantly lower performance compared to other methods, particularly when compared to our proposed V-Laplace and V-Poisson methods.

## B.2 EXPERIMENTS FOR V-GPN ON SEMI-SUPERVISED NODE CLASSIFICATION

The experiments are conducted on two commonly-used citation network datasets (Sen et al., 2008). In the citation networks, the nodes represent documents and their links refer to citations between documents, where each node is associated with a bag-of-words feature vector and a label. Moreover, we use the CE loss with a scale parameter $s$, *i.e.*, $- \log \frac{\exp(s \cdot f(x)_y)}{\sum_{c=1}^{k} \exp(s \cdot f(x)_i)}$ for the labeled node $(x, y)$.

The hyper-parameters, such as learning rate $\eta$, weight decay $wd$, the scale parameter $s$ and the variance-enlarged parameter $\lambda$ are determined by grid search. Concretely, $\eta \in \{0.1, 0.05, 0.01\}$, $wd \in \{1e-2, 5e-3, 1e-3, 5e-4\}$, $s \in \{1, 5, 10\}$, and $\lambda \in \{1.0, 0.5\}$, which can also be found in the code of supplement materials.

Table 7: Classification accuracy (%) of different methods on CIFAR-10 with several labels per class ($\{1, 2, 3, 4, 5\}$). The results described as "mean (std)" are run over 100 trials. Results achieved by V-Laplace and V-Poisson that outperform their original ones are **boldfaced** and the the best results are underlined.

| # LABELS PER CLASS | 1 | 2 | 3 | 4 | 5 |
|---|---|---|---|---|---|
| LAPLACE/LP (ZHU ET AL., 2003) | 10.3 (1.2) | 10.8 (1.7) | 11.8 (2.7) | 13.0 (4.0) | 13.1 (3.4) |
| NEAREST NEIGHBOR | 29.4 (3.9) | 33.4 (3.3) | 35.1 (2.9) | 36.4 (2.3) | 37.4 (2.4) |
| RANDOM WALK (ZHOU & SCHÖLKOPF, 2004) | 37.5 (5.1) | 44.6 (3.5) | 48.4 (3.7) | 51.1 (3.0) | 52.8 (2.8) |
| MBO (GARCIA-CARDONA ET AL., 2014) | 14.2 (4.1) | 19.3 (5.2) | 24.3 (5.6) | 28.5 (5.6) | 33.5 (5.7) |
| WNLL (SHI ET AL., 2017) | 14.9 (4.8) | 24.9 (6.9) | 33.2 (6.7) | 38.4 (7.0) | 42.4 (5.5) |
| CENTERED KERNEL (MAI & COUILLET, 2018) | 35.6 (5.4) | 42.7 (5.4) | 46.0 (3.6) | 48.6 (3.2) | 50.1 (2.7) |
| SPARSE LP (JUNG ET AL., 2016) | 11.8 (2.4) | 12.3 (2.4) | 11.1 (3.3) | 14.4 (3.5) | 11.0 (2.9) |
| P-LAPLACE (RIOS ET AL., 2019) | 34.7 (4.7) | 41.3 (3.5) | 44.6 (3.6) | 47.2 (2.8) | 48.8 (2.8) |
| POISSON (CALDER ET AL., 2020) | 39.1 (5.4) | 45.4 (3.9) | 48.5 (3.6) | 51.2 (3.0) | 52.9 (2.8) |
| V-LAPLACE ($\lambda = 0.001$) | **31.1 (6.0)** | **38.7 (4.7)** | **43.1 (5.3)** | **45.7 (5.0)** | **47.6 (4.3)** |
| V-LAPLACE ($\lambda = 0.002$) | **31.2 (5.9)** | **38.8 (4.6)** | **43.2 (5.3)** | **45.8 (5.0)** | **47.7 (4.3)** |
| V-LAPLACE ($\lambda = 0.005$) | **31.6 (5.9)** | **39.1 (4.6)** | **43.4 (5.2)** | **46.0 (4.9)** | **47.8 (4.2)** |
| V-LAPLACE ($\lambda = 0.01$) | **32.1 (5.9)** | **39.5 (4.5)** | **43.7 (5.1)** | **46.3 (4.9)** | **48.0 (4.2)** |
| V-LAPLACE ($\lambda = 0.02$) | **33.0 (5.7)** | **40.0 (4.4)** | **44.1 (5.0)** | **46.7 (4.8)** | **48.3 (4.2)** |
| V-LAPLACE ($\lambda = 0.05$) | **33.9 (5.6)** | **40.5 (4.3)** | **44.0 (4.9)** | **46.6 (4.8)** | **47.8 (4.3)** |
| V-LAPLACE ($\lambda = 0.1$) | **33.1 (5.6)** | **38.7 (4.6)** | **38.9 (4.7)** | **43.9 (5.3)** | **44.6 (4.9)** |
| V-LAPLACE ($\lambda = 0.2$) | **31.7 (5.5)** | **36.4 (4.5)** | **38.9 (4.7)** | **40.8 (5.4)** | **41.2 (5.0)** |
| V-LAPLACE ($\lambda = 0.5$) | **33.1 (5.4)** | **38.1 (4.5)** | **40.5 (4.6)** | **42.6 (5.2)** | **43.1 (4.8)** |
| V-LAPLACE ($\lambda = 1.0$) | **35.2 (5.7)** | **41.4 (4.3)** | **44.2 (4.7)** | **46.6 (4.8)** | **47.3 (4.1)** |
| V-POISSON ($\lambda = 0.005$) | **39.1 (5.4)** | **46.2 (3.8)** | **49.3 (3.5)** | **52.3 (2.8)** | **54.0 (2.7)** |
| V-POISSON ($\lambda = 0.01$) | **39.2 (5.4)** | **46.3 (3.9)** | **49.3 (3.5)** | **52.3 (2.8)** | **54.0 (2.7)** |
| V-POISSON ($\lambda = 0.05$) | **39.6 (5.5)** | **46.7 (3.9)** | **49.8 (3.5)** | **52.7 (2.8)** | **54.4 (2.7)** |
| V-POISSON ($\lambda = 0.1$) | **40.0 (5.5)** | **47.0 (4.2)** | **50.2 (3.5)** | **53.2 (2.8)** | **54.9 (2.7)** |
| V-POISSON ($\lambda = 0.2$) | **40.3 (5.6)** | **47.1 (3.9)** | **50.8 (3.6)** | **53.9 (2.9)** | **55.5 (2.7)** |
| V-POISSON ($\lambda = 0.5$) | **40.4 (5.5)** | **47.3 (3.9)** | **51.2 (3.6)** | **54.3 (2.9)** | **56.0 (2.8)** |
| V-POISSON ($\lambda = 1.0$) | **41.4 (5.4)** | **48.5 (4.1)** | **51.7 (3.7)** | **54.7 (3.1)** | **56.3 (2.8)** |

Table 8: Classification accuracy (%) of different methods for self-supervised representations on CIFAR-10 with several labels per class ($\{1, 2, 3, 4, 5\}$). The results described as "mean (std)" are run over 100 trials. Results achieved by V-Laplace and V-Poisson that outperform their original ones are **boldfaced** and the the best results are underlined.

| # LABELS PER CLASS | 1 | 2 | 3 | 4 | 5 |
|---|---|---|---|---|---|
| LAPLACE/LP (ZHU ET AL., 2003) | 17.2 (7.0) | 33.4 (10.2) | 49.2 (10.3) | 60.9 (8.9) | 68.8 (7.0) |
| RANDOM WALK (ZHOU & SCHÖLKOPF, 2004) | 69.6 (6.1) | 76.7 (3.5) | 79.4 (3.4) | 81.2 (2.1) | 82.2 (1.9) |
| WNLL (SHI ET AL., 2017) | 52.6 (9.6) | 73.0 (5.5) | 78.7 (4.2) | 81.5 (2.9) | 83.0 (1.9) |
| CENTERED KERNEL (MAI & COUILLET, 2018) | 44.2 (5.2) | 56.1 (4.9) | 63.1 (4.9) | 67.6 (4.3) | 70.5 (3.1) |
| POISSON (CALDER ET AL., 2020) | 73.9 (6.0) | 80.4 (3.0) | 82.7 (2.7) | 84.0 (1.8) | 84.8 (1.6) |
| LINEAR CLASSIFIER | 42.8 (4.3) | 53.9 (4.0) | 60.8 (3.2) | 65.4 (3.3) | 68.8 (2.7) |
| V-LAPLACE | 74.6 (5.7) | 80.9 (2.7) | 83.1 (3.0) | 84.3 (2.0) | 84.8 (1.8) |
| V-POISSON | **75.6 (5.8)** | **81.7 (2.8)** | **83.8 (2.4)** | **84.8 (1.8)** | **85.5 (1.6)** |

