# OpenReview forum: "Variance-enlarged Poisson Learning for Graph-based Semi-Supervised Learning with Extremely Sparse Labeled Data"
_ICLR.cc/2024/Conference — ICLR 2024 poster_

### Official Review · Reviewer_cv8h · 2023-10-27

**Soundness:** 3 good
**Presentation:** 3 good
**Contribution:** 2 fair
**Rating:** 6
**Confidence:** 4

**Summary:**

This paper proposes effective graph-based semi-supervised learning approaches for sparsely labeled data. To improve the accuracy, the proposed approach adds a term of the label variance to the objective function of graph-based semi-supervised learning. The paper conducted experiments to show the effectiveness of the proposed approach.

**Strengths:**

- Graph-based semi-supervised learning is an important research problem in the field.
- The proposed approach is simple and intuitive.
- The theoretical properties of the proposed approach are well discussed in the paper.

**Weaknesses:**

- The compared approaches in the experiment are somewhat old.
- Graph structures used in the experiment are unclear from the description of the paper.

**Questions:**

The paper compares V-Laplace and V-Poisson to other graph-based approaches in the experiment. However, the compared approaches are somewhat old; the most recent one was published in 2020 (POISSON). Similarly, it compares V-GPN to other GNN approaches. However, they are not state-of-the-art, although GNN is a well-studied technique. Is the proposed approach more accurate than recent approaches?

Although k-NN graphs were used in the experiment, the detailed experimental settings are unclear from the descriptions of the paper. k-NN graphs are used in the experiment? What is the number of edges from each node? How do you set edge weight? Is the proposed approach used even if other graph structures are used besides k-NN graphs?

In the datasets used in the experiment, it seems that labels evenly exist. Could you tell me whether the proposed approach is useful for labels of screwed distribution? Please tell me whether the proposed approach is more accurate than other approaches even if labels do not sparsely exist (i.e., we have plenty of labels)? In addition, how do you determine the number of iterations in the proposed approach?

---

> ### Author Response · Authors · 2023-11-19
> **Response to Reviewer cv8h (1/3)**
>
> We thank the reviewer for the constructive comments and insightful suggestions.
>
> **Comment 1**: The paper compares V-Laplace and V-Poisson to other graph-based approaches in the experiment. However, the compared approaches are somewhat old; the most recent one was published in 2020 (POISSON). Similarly, it compares V-GPN to other GNN approaches. However, they are not state-of-the-art, although GNN is a well-studied technique. Is the proposed approach more accurate than recent approaches?
>
> **Response:** Thank you for your valuable comment. We will offer more experimental comparison with recent approaches in the final version. The primary contribution of this work lies in the introduction of a simple yet theoretically sound framework, Variance-Enlarged Poisson Learning (VPL). This framework is designed to tackle the issue of degenerate solutions, which tend to be nearly constant for unlabeled points. Our work is poised to offer significant value to the ICLR community by providing novel and impactful insights. Additionally, we present two specific algorithms, namely V-Laplace and V-Poisson. As they are advanced versions of the original Laplace learning and Poisson learning, we choose to demonstrate their superiority through experimental comparisons with variants of Laplace learning and Poisson learning.
>
> ------
>
> **Comment 2:** "Graph structures used in the experiment are unclear from the description of the paper." and "Although k-NN graphs were used in the experiment, the detailed experimental settings are unclear from the descriptions of the paper. k-NN graphs are used in the experiment? What is the number of edges from each node? How do you set edge weight? Is the proposed approach used even if other graph structures are used besides k-NN graphs?"
>
> **Response:** Thank you for your kind comment.
> Due to page limitations, we present detailed information on k-NN graph construction in Appendix B. Specifically, for each dataset, we generate a graph in the latent feature space using all available data, resulting in $n=70,000$ nodes for MNIST and FashionMNIST, and $n=60,000$ nodes for CIFAR-10. The graph is constructed as a $K$-nearest neighbor graph with edge weights calculated using the formula $w_{ij}=\exp(-4\lVert x_i-x_j\Vert^2/d_K(x_i)^2)+\epsilon$, where $x_i$ denotes the latent variables for sample $i$, $d_K(x_i)$ represents the distance in the latent space between $x_i$ and its K-th nearest neighbor, and $\epsilon\ge 0$.
>
> ------
>
> **Comment 3:** In the datasets used in the experiment, it seems that labels evenly exist. Could you tell me whether the proposed approach is useful for labels of screwed distribution? Please tell me whether the proposed approach is more accurate than other approaches even if labels do not sparsely exist (i.e., we have plenty of labels)? In addition, how do you determine the number of iterations in the proposed approach?
>
> **Response:** Thank you for your insightful comments and inquiries. We have taken them into careful consideration and conducted additional experiments to address your concerns.
>
> `About labels with screwed distribution`
> To assess the effectiveness of our proposed approach on labels with skewed distribution, we conduct experiments specifically on labeled data characterized by a long-tailed distribution. In this scenario, the distribution of labels follows a long-tailed pattern, with the first class having $n_1$ labels and subsequent classes having progressively more labels, that is, the $i$-th class is randomly assigned $n_i=\lfloor {n_1}/{\eta^{\frac{i-1}{k-1}}}\rfloor$ labels, where $k$ represents the number of classes, and $\eta$ denotes the imbalance factor. The experimental results on MNIST with imbalance ratios $\eta\in{0.1, 0.2, 0.5}$ are summarized in the tables below. These results demonstrate the effectiveness of our proposed approaches, V-Laplace and V-Poisson, in achieving higher accuracy compared to other methods when dealing with labels of skewed distribution.
>
> Imbalance Ratio=0.1
>
> | \# Labels per Class |       1        |       2        |       3        |     4      |     5      |
> | :----------------: | :------------: | :------------: | :------------: | :--------: | :--------: |
> |     Laplace/LP     |   11.5 (2.8)   |   23.5 (5.4)   |   35.3 (2.7)   | 40.0 (1.8) | 42.7 (1.9) |
> |    Random Walk     |   77.4 (3.6)   |   85.5 (2.2)   |   88.1 (1.9)   | 90.4 (1.2) | 91.0 (1.1) |
> |        WNLL        |   46.3 (4.2)   |   74.6 (6.4)   |   89.2 (3.3)   | 94.2 (1.2) | 95.3 (0.7) |
> |   Centerd Kernel   |   82.2 (3.2)   |   89.4 (1.9)   |   91.4 (1.5)   | 93.6 (1.2) | 94.2 (0.9) |
> |      Poisson       |   89.5 (2.8)   |   93.5 (1.4)   |   94.5 (1.1)   | 95.2 (0.5) | 95.4 (0.4) |
> |   **V-Laplace**    |   89.6 (4.1)   | **93.7 (1.9)** |   94.5 (1.5)   | 95.2 (0.5) | 95.1 (0.5) |
> |   **V-Poisson**    | **89.7 (3.6)** |   93.5 (1.7)   | **94.6 (1.2)** | 95.2 (0.5) | 95.4 (0.4) |

---

> ### Author Response · Authors · 2023-11-19
> **Response to Reviewer cv8h (2/3)**
>
> Imbalance Ratio=0.2
>
> | \# Labels per Class |       1        |       2        |       3        |       4        |       5        |
> | :----------------: | :------------: | :------------: | :------------: | :------------: | :------------: |
> |     Laplace/LP     |   13.3 (4.0)   |   18.3 (5.7)   |   31.2 (6.3)   |   38.0 (4.0)   |   43.1 (2.9)   |
> |    Random Walk     |   81.3 (3.8)   |   88.4 (2.0)   |   91.5 (1.6)   |   92.5 (1.3)   |   93.4 (0.9)   |
> |        WNLL        |   45.3 (6.2)   |   74.4 (5.7)   |   91.4 (2.7)   |   93.9 (1.7)   |   95.7 (0.7)   |
> |   Centerd Kernel   |   84.0 (4.0)   |   90.1 (1.7)   |   92.5 (1.5)   |   93.6 (1.3)   |   94.3 (1.0)   |
> |      Poisson       |   90.8 (3.0)   |   94.5 (1.1)   |   95.4 (0.9)   |   95.8 (0.6)   |   96.0 (0.5)   |
> |     V-Lapalce      | **91.9 (3.8)** | **95.4 (0.9)** | **95.9 (0.7)** |   96.1 (0.4)   |   96.2 (0.2)   |
> |     V-Poisson      |   91.3 (3.6)   |   95.2 (1.1)   |   95.8 (1.0)   | **96.1 (0.6)** | **96.3 (0.4)** |
>
> Imbalance Ratio=0.5
>
> | \# Labels per Class |       1        |       2        |       3        |       4        |       5        |
> | :----------------: | :------------: | :------------: | :------------: | :------------: | :------------: |
> |     Laplace/LP     |   13.5 (5.5)   |   21.3 (9.9)   |  32.9 (13.3)   |  46.2 (13.4)   |  54.3 (11.7)   |
> |    Random Walk     |   83.9 (4.7)   |   90.5 (2.2)   |   92.7 (1.6)   |   93.9 (1.2)   |   94.6 (0.7)   |
> |        WNLL        |  41.2 (14.2)   |   80.1 (7.8)   |   92.4 (3.3)   |   95.3 (1.0)   |   95.9 (0.6)   |
> |   Centerd Kernel   |   84.2 (4.9)   |   90.9 (2.0)   |   92.9 (1.7)   |   94.1 (1.2)   |   94.3 (1.0)   |
> |      Poisson       |   90.8 (4.0)   |   94.4 (1.3)   |   95.4 (0.9)   |   95.8 (0.8)   |   96.1 (0.4)   |
> |     V-Lapalce      |   91.7 (3.8)   |   95.2 (1.0)   |   96.0 (0.7)   |   96.2 (0.6)   |   96.5 (0.4)   |
> |     V-Poisson      | **92.2 (3.5)** | **96.0 (0.7)** | **96.4 (0.5)** | **96.5 (0.5)** | **96.7 (0.2)** |
>
> ------
>
> `About more labels`
>
> In response to your query regarding experiments with more labels, we have conducted experiments to investigate the performance of our proposed V-Laplace and V-Poisson in scenarios where the number of labels is increased. The results, presented in the table below, indicate that as the number of labels increases, the performance gain of our proposed methods diminishes and becomes weaker compared to Laplace and Random Walk methods, but it is still slightly better than Poisson learning.
>
> MNIST
>
> | \# Labels per Class |       10       |       20       |       40       |       80       |      160       |
> | :----------------: | :------------: | :------------: | :------------: | :------------: | :------------: |
> |     Laplace/LP     |   94.5 (2.4)   |   96.6 (0.2)   |   96.9 (0.1)   | **97.1 (0.1)** | **97.3 (0.1)** |
> |    Random Walk     |   95.6 (0.4)   |   96.3 (0.2)   |   96.7 (0.1)   |   97.0 (0.1)   |   97.2 (0.1)   |
> |        WNLL        |   96.6 (0.3)   |   96.8 (0.2)   |   96.9 (0.1)   |   96.9 (0.1)   |   96.8 (0.1)   |
> |   Centerd Kernel   |   95.4 (0.7)   |   96.2 (0.2)   |   96.6 (0.1)   |   96.8 (0.1)   |   97.0 (0.1)   |
> |      Poisson       |   96.5 (0.2)   |   96.7 (0.1)   |   96.8 (0.1)   |   96.9 (0.1)   |   96.9 (0.1)   |
> |     V-Lapalce      |   96.7 (0.3)   | **96.9 (0.1)** |   97.0 (0.1)   |   97.0 (0.1)   |   97.0 (0.0)   |
> |     V-Poisson      | **96.7 (0.2)** | **96.9 (0.1)** | **97.1 (0.1)** | **97.1 (0.1)** |   97.1 (0.0)   |
>
> FashionMNIST
>
> | \# Labels per Class |       10       |       20       |       40       |       80       |      160       |
> | :----------------: | :------------: | :------------: | :------------: | :------------: | :------------: |
> |     Laplace/LP     |   70.9 (2.4)   |   76.1 (1.3)   |   79.0 (0.7)   |   80.7 (0.5)   |   82.1 (0.3)   |
> |    Random Walk     |   73.8 (1.4)   |   77.0 (0.9)   |   79.3 (0.6)   |   80.9 (0.4)   | **82.2 (0.3)** |
> |        WNLL        |   74.0 (1.7)   |   77.4 (0.9)   |   79.5 (0.6)   |   80.8 (0.4)   |   81.6 (0.3)   |
> |   Centerd Kernel   |   68.3 (2.0)   |   73.8 (1.2)   |   77.0 (0.6)   |   79.2 (0.4)   |   80.9 (0.3)   |
> |      Poisson       |   74.2 (1.5)   |   76.7 (1.1)   |   78.2 (0.8)   |   79.2 (0.7)   |   80.0 (0.5)   |
> |     V-Lapalce      |   75.4 (1.4)   |   77.7 (0.9)   |   79.1 (0.9)   |   80.4 (0.4)   |   80.1 (0.4)   |
> |     V-Poisson      | **75.7 (1.4)** | **78.2 (0.9)** | **79.9 (0.8)** | **81.2 (0.5)** |  81.7  (0.4)   |
>
> ------

---

> ### Author Response · Authors · 2023-11-19
> **Response to Reviewer cv8h (3/3)**
>
> `About Iteration Termination`
> Regarding the determination of the number of iterations in our proposed approach, we employ a technique inspired by Calder et al. (2020). Specifically, we incorporate the iteration step $p_{t+1}=WD^{-1}p_t$ in the algorithms of V-Laplace and V-Poisson. The initial vector $p_0$ is initialized with ones at the positions of all labeled vertices and zeros elsewhere. The iterations for V-Laplace and V-Poisson are terminated once the infinity norm of the difference between $p_t$ and $p_{\infty}$, i.e., $\Vert p_t-p_{\infty}\Vert_{\infty}$, is less than or equal to $\frac{1}{n}$. This termination criterion is chosen just before reaching the mixing time. Herein, $p_{\infty}=W1 \frac{1}{1^\top W1}$.

---

> ### Author Response · Authors · 2023-11-23
> **Looking forward to your feedback**
>
> Dear reviewer cv8h,
>
> Thanks again for your valuable time and insightful comments. As the deadline for the Author/Reviewer discussion is approaching, it would be nice of you to let us know whether our answers have solved your concerns so that we can better improve our work. We are looking forward to your feedback.
>
> Best regards,
>
> Authors of Paper 221

---

### Official Review · Reviewer_JXpL · 2023-10-28

**Soundness:** 3 good
**Presentation:** 4 excellent
**Contribution:** 3 good
**Rating:** 6
**Confidence:** 4

**Summary:**

The paper introduces a graph-based transductive semi-supervised learning method, which modifies Laplace and Poisson learning techniques by incorporating a variance-enlarged term for regularization. The authors present algorithms for these 'variance-enlarged' learning methods. Additionally, they propose a novel message passing layer with attention for Graph Neural Networks (GNNs) to enhance label variance, based on the 'label propagation' step of their algorithms. These contributions are tested in scenarios with limited labeled data and compare favorably against other methods.

On the theoretical front, the paper explores both discrete and variational cases. In the discrete case, the variance enlarged approach corresponds to a reduction of the edge-weights, which, under certain conditions, strengthens connections between nodes of the same class and weakens those between nodes of different classes. In the variational case, the minimizer of theoptimization problem is expressed as the solution of a PDE.

**Strengths:**

- The paper is well written.

- The paper presents a variety of contributions of theoretical and practical nature.

- The proposed algorithms are simple yet effective as shown in the experiments section.

**Weaknesses:**

I suspect Lemma 4.1 could have an error and therefore all proofs that derive from it. See questions.

**Questions:**

1. **Iterative vs. Linear Solution:**

   - In Algorithm 1, you've chosen an iterative approach for solving V-Laplace Learning. However, the conventional Laplace learning approach can be solved directly through a linear system ([1,2]). I'm curious if your method could also utilize a linear system solution. Is there a specific reason for the iterative approach? Is it faster?

2. **Relation to Previous Work:**

   - In [2], Laplace learning was associated with the probability of sampling a spanning forest that separates the seeds. Do you think your approach could also have a similar interpretation in this context?

3. **Consideration of Directed Graphs:**

   - As far as I understand, your approach does not consider directed graphs. Does your approach and the theoretical insights extend to the directed case as well?

4. **Convergence Dependency on Parameters:**

   - I'm interested in understanding how the convergence of your algorithms is influenced by the value of $\lambda$. Could you shed some light on this relationship?

5. **Clarification on Lemma 4.1:**

   - In the final step of Lemma 4.1, it seems that the sum of $q_j$ is factored out of the norm. However, this step isn't clear to me. Could you provide a more explicit explanation of how this is done? To illustrate, if I consider $(u_1,u_2)=(0.5,1)$, $(q_1,q_2)=(0.5,0.5)$, $\lambda=1$ and $w_{12}=w_{21}=1$, the equation doesn't seem to balance. The right term of the inequality is equal to
$\sum_i^n\sum_j^n(w_{ij}-\lambda q_iq_j)||u(x_i)-u(x_j)||_2^2=$

$=2\left((w_{12}-\lambda q_1q_2)(u_1-u_2)^2\right)=2\left((1-0.5^2)(0.5-1)^2\right)=0.375$

While the left term is equal to

$\sum_i^n\sum_j^n(w_{ij})||u(x_i)-u(x_j)||_2^2-\lambda\sum_i^nq_i\left|\left|u(x_i)-\sum_j^nq_ju(x_j)\right|\right|^2_2=$

$=2\left(w_{12}(u_1-u_2)^2\right)-\left(q_1\left(u_1-\left(q_1u_1+q_2u_2\right)\right)^2+q_2\left(u_2-\left(q_1u_1+q_2u_2\right)\right)^2\right)=$
$=2\cdot0.5^2-\left(0.5\left(0.5-\left(0.5\cdot0.5+0.5\cdot 1\right)\right)^2+0.5\left(1-\left(0.5\cdot0.5+0.5\cdot 1\right)\right)^2\right)\\
     =0.4375$

This clarification is crucial as Theorem 3.1 and Proposition 4.3 depend on this Lemma.

6. **Typos**:

   - In proposition 4.3 references Theorem 4.1. It should be Lemma 4.1.

   - Table 7  does not contain the accuracies for the V-Poisson method.

   - Table 5 and 6 do not contain any clarification regarding the meaning of the bold values.

[1] Grady, "Random Walks for Image Segmentation" (2006)

[2] Fita Sanmartin et al. “Probabilistic Watershed: Sampling all spanning forests for seeded segmentation and semi-supervised learning” (2019)

---

> ### Author Response · Authors · 2023-11-19
> **Response to Reviewer JXpL (1/2)**
>
> We thank the reviewer for the constructive comments and insightful suggestions.
>
> `About Iterative vs. Linear Solution`
>
> **Comment:** In Algorithm 1, you've chosen an iterative approach for solving V-Laplace Learning. However, the conventional Laplace learning approach can be solved directly through a linear system ([1,2]). I'm curious if your method could also utilize a linear system solution. Is there a specific reason for the iterative approach? Is it faster?
>
> **Response:** Thank you for your insightful comment. In Section 3.2, we introduce V-Laplace (Eq. 3.3) and V-Poisson (Eq. 3.4) with linear equations. Indeed, as suggested by the reviewer, these equations can be solved using classical methods for linear systems. However, it is crucial to highlight that solving V-Laplace and V-Poisson through direct methods entails high complexity. This involves the inverse or pseudoinverse process of a dense matrix associated with $W' = W - \lambda qq^\top \in \mathbb{R}^{n \times n}$ (where $n=70000$ for MNIST and Fashion MNIST, and $n=60000$ for CIFAR datasets), resulting in a computational complexity of $O(n^3)$.
>
> Therefore, we adopt an iterative approach that involves only matrix multiplication, offering a lower computational complexity of $O(Tn^2)$, where $T$ usually takes between 100 and 200. Additionally, for the sake of efficiency, we do not directly use the dense matrix $W'$ to compute the Laplacian matrix during iterations. This is motivated by the sparsity of $W$ in our experiments, involving k-NN graphs and masked attention maps, enabling us to leverage sparse matrix multiplication for substantial computational speedup. The overall iteration time is about 1-3 seconds.
>
> ------
>
> `About Relation to Previous Work`
>
> **Comment:** In [2], Laplace learning was associated with the probability of sampling a spanning forest that separates the seeds. Do you think your approach could also have a similar interpretation in this context?
>
> **Response**: Thank you for providing additional insights into the connection between our approach and Laplace learning. We appreciate your thoughtful comment and the reference to Section 3.1 of [2], where Laplace learning is associated with the probability of sampling a spanning forest that separates the seeds.
>
> Upon further consideration, we acknowledge that while our approach may not have a directly similar interpretation to the probability of sampling spanning forests as in Laplace learning, but there is a potential connection through the introduction of variance enlargement. Specifically, Lemma 4.1 in our work introduces variance enlargement, which, akin to Laplace learning, has the effect of reducing the edge weights in the graph. In this context, the reduction of edge weights can be seen as a way of decreasing the weight $w(f)$ of a forest in the graph, thus amplifying the importance of forests with large weights (similar to Proposition 4.3 in our work). This somewhat coincides with that the lower energy can give the 2-forests of lowest cost more probability mass $p(f)$.
>
> -------
>
> `About Consideration of Directed Graphs`
>
> **Comment:** As far as I understand, your approach does not consider directed graphs. Does your approach and the theoretical insights extend to the directed case as well?
>
> **Response:** Thank you for your thoughtful comment. Our proposed algorithms, V-Laplace and V-Poisson, can be effectively applied to directed graphs. This versatility arises from the fact that the Poisson equations in Eq. 3.2 and Eq. 3.3 do not mandate the weight matrix $W$ to be symmetric. It is noteworthy that when working with directed graphs, considerations may arise regarding whether to utilize the in-degree matrix or out-degree matrix when defining the Laplacian matrix.
>
> The variational analysis detailed in Section 4.2 cannot be extended to the directed case, because the theoretical results in this section are established under the assumption of asymptotic limits, particularly as the bandwidth of the symmetric edge weight $w_{ij}=\psi\left(\frac{\Vert x_i-x_j\Vert_2^2}{h}\right)$ tends to zero.
>
> ------

---

> ### Author Response · Authors · 2023-11-19
> **Response to Reviewer JXpL (2/2)**
>
> `About Convergence Dependency on Parameters`
>
> **Comment:** I'm interested in understanding how the convergence of your algorithms is influenced by the value of $\lambda$. Could you shed some light on this relationship?
>
> **Response:** Thanks for your valuable comment. We would like to provide analysis about the influence of $\lambda$ to convergence.  In term of the total number of iterations, we employ the trick by Calder et al. (2020). Specifically, we add the iteration step $p_{t+1}=WD^{-1}p_t$ in the algorithms of V-Laplace and V-Poisson,  where $p_0$ is initialized as a vector with ones at the positions of all labeled vertices and zeros  elsewhere. The iterations for V-Laplace and V-Poisson are terminated once $\Vert p_t-p_{\infty}\Vert_{\infty}\le \frac{1}{n}$ is achieved, just before reaching the mixing time. Herein, $p_{\infty}=W 1/(1^\top W1)$. Therefore, the algorithm's iteration is solely tied to the specific graph and remains independent of $\lambda$.
>
> ------
>
> `About Clarification on Lemma 4.1`
>
> **Comment:** I suspect Lemma 4.1 could have an error and therefore all proofs that derive from it. See questions.
>
> **Response:** Thanks very much for your kind reminding! We appreciate your careful review. We have carefully examined Lemma 4.1 and made revision about it in the manuscript.  Please refer to the highlighted part in the updated Lemma 4.1.
>
> The corrected expression for the distance measure is $L(u(x_i),u(x_j))=\frac{1}{2}\Vert u(x_i)-u(x_j)\Vert_2^2$ in Eq. 3.2. Consequently, the equivalent objective should be $\frac{1}{2}\sum_{i=1}^n\sum_{j=1}^n (w_{ij}-\lambda q_iq_j)\Vert u(x_i)-u(x_j)\Vert_2^2$ since we have
> \begin{equation}
> \begin{aligned}
> \mathrm{Var}[u]=&\sum_{i=1}^n q_i\Vert u(x_i)-\overline{u}\Vert_2^2\\\\
> =&\sum_{i=1}^n q_i\Vert u(x_i)\Vert_2^2-\sum_{i=1}^n\sum_{j=1}^n q_iq_j\langle u(x_i),u(x_j)\rangle\\\\
> =&\frac{1}{2} \sum_{i=1}^n\sum_{j=1}^n q_iq_j(\Vert u(x_i)\Vert_2+\Vert u(x_j)\Vert_2^2) - \sum_{i=1}^n\sum_{j=1}^n q_iq_j\langle u(x_i),u(x_j)\rangle\\\\
> =&\frac{1}{2}\sum_{i=1}^n\sum_{j=1}^n q_iq_j \Vert u(x_i)-u(x_j)\Vert_2^2.
> \end{aligned}
> \end{equation}
>
> Concerning the provided illustration, we have $\frac{1}{2}\sum_{i=1}^n\sum_{j=1}^n (w_{ij}-\lambda q_i q_j)\Vert u(x_i)-u(x_j)\Vert_2^2=0.1875$ and $\frac{1}{2}\sum_{i=1}^n\sum_{j=1}^n w_{ij}\Vert u(x_i)-u(x_j)\Vert_2^2-\lambda \mathrm{Var}[u]=0.5^2-0.0625=0.1875$. It's important to note that the correction made to $L$ does not alter the corresponding conclusions in Theorem 3.1 and Proposition 4.3 based on this lemma.

---

> > ### Comment · Reviewer_JXpL · 2023-11-22
> >
> > Thank you for your response. Since my concerns have been addressed, I will raise my score.

---

> > > ### Author Response · Authors · 2023-11-23
> > > **Thanks for your feedback.**
> > >
> > > Dear Reviewer JXpL,
> > >
> > > Thank you for your feedback! We are pleased to address your concerns and greatly appreciate your reviews, which play a crucial role in improving our work.
> > >
> > > Best regards,
> > > Authors of Paper 221

---

### Official Review · Reviewer_CApx · 2023-10-31

**Soundness:** 3 good
**Presentation:** 3 good
**Contribution:** 3 good
**Rating:** 6
**Confidence:** 5

**Summary:**

This paper addresses the issue of traditional graph-based semi-supervised learning leading to degenerate solutions when labeled data is extremely sparse. It introduces VPL, which mitigates this problem by increasing the variance of predictions for unlabeled data. Furthermore, based on classical Laplace learning and Poisson learning, the paper proposes V-Laplace and V-Poisson as improvements. Extensive experiments have demonstrated the effectiveness of these approaches.

**Strengths:**

- This paper provides an overview of classical graph-based semi-supervised learning tasks, and the proposed idea, while simple, is highly effective. Its parameter-free nature makes it more appealing.
- The writing in this paper is of excellent quality, and the motivation and introduction of the proposed method are presented in a clear and easily understandable manner.
- This paper provides a thorough and reliable theoretical analysis.
- In addition to the general graph-based SSL methods like Laplace Learning, this paper also extends to GNN-based method and proposes V-GPN.

**Weaknesses:**

- Disclaimer: I am familiar with GNN-based semi-supervised learning and have knowledge of Laplace Learning and Poisson Learning, but I am not familiar with their applications in non-graph structured data. I noticed that the experiments primarily focus on datasets like (Fashion) MNIST and CIFAR-10. It would be beneficial to expand the experiments to larger datasets, such as ImageNet.
- Typos: e.g., lambda in the caption of Figure 1.

My other concern is the practical value of such graph-based (parameter-free) semi-supervised learning methods. As shown in Table 2 and Table 3, despite significant improvements over the baselines, the accuracy of the proposed method still falls short of being satisfactory. To my knowledge, parameterized models like ResNet and ViT-based self-supervised learning methods tend to perform better in cases of label sparsity. Therefore, in resource-abundant scenarios, it seems that having a parameter-free model with relatively poorer performance may not be very meaningful.

**Questions:**

I hope the authors can answer the last point in Weaknesses, and I am glad to raise my score if my concern can be addressed.

---

> ### Author Response · Authors · 2023-11-19
> **Response to Reviewer CApx (1/2)**
>
> We thank the reviewer for the constructive comments and insightful suggestions.
>
> **Comment 1:** Disclaimer: I am familiar with GNN-based semi-supervised learning and have knowledge of Laplace Learning and Poisson Learning, but I am not familiar with their applications in non-graph structured data. I noticed that the experiments primarily focus on datasets like (Fashion) MNIST and CIFAR-10. It would be beneficial to expand the experiments to larger datasets, such as ImageNet.
>
> **Response:** Thanks very much for your kind comment. Your suggestion to extend our experiments to include larger datasets, particularly ImageNet, is certainly noteworthy. However, it is imperative to acknowledge the inherent challenges associated with such an expansion, especially given the extensive scale of ImageNet-1K. The graph structure within ImageNet-1K is notably vast, boasting a node count of 1.2 million. Furthermore, the intricacies introduced by the graph weight matrix, with dimensions on the order of $10^6\times 10^6$, pose formidable computational and methodological challenges. Addressing such challenges requires careful consideration and exploration of alternative strategies. While the direct application of our methods to ImageNet-1K may pose difficulties, we are actively exploring avenues to enhance scalability and extend the applicability to larger datasets.
>
> ------
>
> **Comment 2:** My other concern is the practical value of such graph-based (parameter-free) semi-supervised learning methods. As shown in Table 2 and Table 3, despite significant improvements over the baselines, the accuracy of the proposed method still falls short of being satisfactory. To my knowledge, parameterized models like ResNet and ViT-based self-supervised learning methods tend to perform better in cases of label sparsity. Therefore, in resource-abundant scenarios, it seems that having a parameter-free model with relatively poorer performance may not be very meaningful.
>
> **Response:** Thank you for your insightful comment. The focus of our paper is classification, a task where two components play a crucial role in determining the overall performance: representation learning and classifier learning. Although self-supervised learning has demonstrated promising results in representation learning, it encounters challenges in effectively learning classifiers, especially in scenarios marked by severely restricted labeled data, such as situations where there is only one label per class.
>
> To illustrate this point, we refer to the open-source code and model parameters associated with an exemplary linear probe achievement of 91.9\% on CIFAR-10. The source code and model parameters are available at [SimSiam-on-CIFAR10](https://github.com/Reza-Safdari/SimSiam-91.9-top1-acc-on-CIFAR10), which utilizes the SimSiam framework. Building upon the pretrained model, we conduct two experiments to underscore the effectiveness of graph-based semi-supervised learning in scenarios of label scarcity: (i) We utilize the SimSiam pretrained model to extract features from the entire CIFAR-10 dataset and construct a 10-nearest-neighbors graph on these representations. We then randomly choose which data points are labeled, and predict unlabeled data for all data points using different graph-based semi-supervised learning approaches; (ii) For comparison, we directly use several randomly chosen labeled data to obtain a linear classifier, and finally use this classifier to evaluate the classification accuracy of all samples.
>
> The experimental results are summarized in the table below. It is evident that the linear classifier, trained solely on labeled data, outperforms Laplace learning in scenarios with extremely sparse data. However, it is crucial to note that linear classification exhibits significantly lower performance compared to other methods, particularly when compared to our proposed V-Laplace and V-Poisson methods. Also, it is worth noting that V-Poisson only needs 5 labels to achieve a good result of 85.5 (1.6). This demonstrates the effectiveness and potential of graph-based semi-supervised learning. Many thanks to the reviewers for their thoughts about self-supervised learning and this work, we have added the experiments to the newly revised version.

---

> > ### Comment · Reviewer_CApx · 2023-11-23
> > **Response to the Author**
> >
> > Thank you for the response. I am glad to see the new experiments about the comparison with self-supervised learning methods. I wish to see the extension of the proposed method to larger-scale datasets. I will keep the score but raise the confidence.

---

> > > ### Author Response · Authors · 2023-11-23
> > > **Many thanks for your feedback!**
> > >
> > > Many thanks to the reviewer for your approval of the new experiments. We will enhance our research by incorporating more experiments on large-scale datasets in the future version. Thank you once again.
> > >
> > > Best regards,
> > > The authors

---

> ### Author Response · Authors · 2023-11-19
> **Response to Reviewer CApx (2/2)**
>
> | # Labels per Class | 1              | 2              | 3              | 4              | 5              |
> | :-------------------------: | -------------- | -------------- | -------------- | -------------- | -------------- |
> |         Laplace/LP          | 17.2 (7.0)     | 33.4 (10.2)    | 49.2 (10.3)    | 60.9 (8.9)     | 68.8 (7.0)     |
> |         Random Walk         | 69.6 (6.1)     | 76.7 (3.5)     | 79.4 (3.4)     | 81.2 (2.1)     | 82.2 (1.9)     |
> |            WNLL             | 52.6 (9.6)     | 73.0 (5.5)     | 78.7 (4.2)     | 81.5 (2.9)     | 83.0 (1.9)     |
> |       Centerd Kernel        | 44.2 (5.2)     | 56.1 (4.9)     | 63.1 (4.9)     | 67.6 (4.3)     | 70.5 (3.1)     |
> |           Poisson           | 74.5 (6.2)     | 81.0 (3.0)     | 83.0 (2.7)     | 84.1 (1.9)     | 84.8 (1.6)     |
> |    **Linear Classifier**    | 42.8 (4.3)     | 53.9 (4.0)     | 60.8 (3.2)     | 65.4 (3.3)     | 68.8 (2.7)     |
> |        **V-Laplace**        | 74.6 (5.7)     | 80.9 (2.7)     | 83.1 (3.0)     | 84.3 (2.0)     | 84.8 (1.8)     |
> |        **V-Poisson**        | **75.6 (5.8)** | **81.7 (2.8)** | **83.8 (2.4)** | **84.8 (1.8)** | **85.5 (1.6)** |
>
> ------
>
>
> **Comment 3:** Typos: e.g., lambda in the caption of Figure 1.
>
> **Response**: Thank you for your careful review. We have rectified the mentioned typos and address some other potential errors.

---

> ### Author Response · Authors · 2023-11-23
> **Looking forward to your feedback**
>
> Dear reviewer CApx,
>
> Thanks again for your valuable time and insightful comments. As the deadline for the Author/Reviewer discussion is approaching, it would be nice of you to let us know whether our answers have solved your concerns so that we can better improve our work. We are looking forward to your feedback.
>
> Best regards,
>
> Authors of Paper 221

---

### Official Review · Reviewer_HipA · 2023-11-05

**Soundness:** 3 good
**Presentation:** 3 good
**Contribution:** 3 good
**Rating:** 6
**Confidence:** 3

**Summary:**

Semi-supervised learning (SSL) aims to leverage a vast amount of freely available unlabeled data alongside a small sample of expensive labeled data to improve the classification performance of a learnt model. Graph SSL techniques are a popular class of approaches where by constructing a graph with data points nodes and relationship edges, information can be propagated across the graph to make predictions on unlabeled data. A well-recognized limitation of typical Graph SSL techniques is the problem of degenerate solutions where, when the labeled sample is sparse, the nodes far away from any labeled sample can converge to a constant and uninformative value.

This paper proposes a simple and intuitive fix to degeneracy issue by regularizing the node values to be different from one another through a term that increases the variance between node values. Clear theoretical insights have been provided to show that, when the graph edges connect same class nodes more often than others, variance enlargement can amplify the importance of edge weights connecting vertices within the same class, while simultaneously diminishing the importance of those connecting vertices from different classes, thus leading to improved solutions.

Experimental results show salient gains due to variance enlargement regularize on a variety of datasets.

**Strengths:**

* SSL is an important problem and the paper addresses the crucial issue of node degeneracy in Graph SSL. As such, the problem is well-motivated
* The solution is simple and intuitive and theoretical connections are provided to explain the inner workings of the proposed technique
* Experiments are conducted on a wide range of datasets and show significant gains which demonstrates the utility of the technique

**Weaknesses:**

* Many variants of Graph SSL have been proposed in the literature. It will be interesting to discuss the effect of variance enlargement on those also beyond V-GPN that the paper explores.

**Questions:**

Can variance enlargement help with other Graph SSL approaches besides GPN? If so, a discussion on where and why it helps and does not help, can be useful and interesting.

---

> ### Author Response · Authors · 2023-11-19
> **Response to Reviewer HipA**
>
> We thank the reviewer for the constructive comments and insightful suggestions.
>
> **Comment**: *"Many variants of Graph SSL have been proposed in the literature. It will be interesting to discuss the effect of variance enlargement on those also beyond V-GPN that the paper explores."* and *"Can variance enlargement help with other Graph SSL approaches besides GPN? If so, a discussion on where and why it helps and does not help, can be useful and interesting."*
>
> **Response**: Thank you for your insightful comment. We appreciate the opportunity to consider the broader applicability of variance enlargement to other Graph SSL approaches. Upon a careful examination of Algorithm 1 and Algorithm 2, it becomes evident that variance enlargement corresponds to a straightforward step—specifically, the incorporation of $\lambda\overline{U}$ into the message passing process. To illustrate, within the framework of Graph Convolutional Networks (GCN), the layer-wise propagation rule is typically defined as $H^{(l+1)}=\sigma(D^{-1/2}AD^{-1/2} H^{(l)}W^{(l)})$, where $D^{-1/2}AD^{-1/2} (H^{(l)}W^{(l)})$ indicates the propagation of the message $H^{(l)}W^{(l)}$. Analogous to V-Laplace and V-Poisson, the incorporation of variance enlargement directly enlarges the variance of $H^{(l)}W^{(l)}$ into the graph convolutions:
> $$
> H^{(l+1)}=\sigma(D^{-1/2}AD^{-1/2} H^{(l)}W^{(l)} + \lambda \overline{U}),
> $$
> where $\overline{U}=\overline{H^{(l)}W^{(l)}}$ denotes the (weighted) zero-mean normalization of $H^{(l)}W^{(l)}$. This incorporation can be generalized to various Graph SSL approaches, as we can treat variance enlargement as a step in the aggregation process of message-passing neural networks [1], thereby enhancing the dispersion of messages.
>
> Thanks again for your kind suggestion, which has inspired us to delve into the application of variance enlargement across a spectrum of Graph SSL approaches, extending beyond V-GPN. In our forthcoming research, we aim to broaden our investigation, providing a thorough discussion on the nuanced aspects of where and why variance enlargement demonstrates its effectiveness or limitations within diverse Graph SSL methodologies. This endeavor is intended to offer a comprehensive understanding of the implications of variance enlargement, contributing valuable insights to the field and fostering a deeper comprehension of its applicability and impact across a diverse range of Graph SSL frameworks.
>
>
> > [1] Xu K, Hu W, Leskovec J, et al. How Powerful are Graph Neural Networks?[C]//International Conference on Learning Representations. 2018.

---

> > ### Comment · Reviewer_HipA · 2023-11-22
> >
> > I thank the authors for their insights. I keep my score.

---

> > > ### Author Response · Authors · 2023-11-23
> > > **Thank you!**
> > >
> > > Dear Reviewer HipA,
> > >
> > > Thank you for your feedback! We are pleased to address your concerns and greatly appreciate your reviews, which play a crucial role in improving our work.
> > >
> > > Best regards,
> > >
> > > Authors of Paper 221

---

### Author Response · Authors · 2023-11-23
**Appreciation for Your Review and Openness to Further Comments**

Dear Reviewer,

We sincerely appreciate your dedicated time and effort in reviewing our paper. Your insightful comments have been invaluable in enhancing the quality of our work. Should you have any additional comments or queries regarding our response, we would be delighted to address them.

Thank you once again for your thorough review.

Best regards,

The Authors

---

### Meta-Review · Area_Chair_JTgB · 2023-12-10

**Metareview:**

This paper addresses the issue of degenerate solutions occurring during semisupervised learning on graphs by introducing a term to encourage increased variance across the node values. While the reviewers point out that it would strengthen the authors' claims to show the results of employing the same technique to other graph SSL techniques, they agreed that the paper provides a simple solution to an important problem, the theory given provides valuable insight into the performance of the method, and that the experimental section demonstrates convincing performance gains when the method is used. For these reasons, acceptance is recommended.

**Justification For Why Not Higher Score:**

The competitiveness of the proposed semi-supervised method is limited to the very sparse case.

**Justification For Why Not Lower Score:**

The paper makes an interesting and empirically useful technical contribution.

---

### Decision · Program_Chairs · 2024-01-16

Accept (poster)